# Modelling the effect of aerosol and greenhouse gas forcing on the South & East Asian monsoons with an intermediate complexity climate model

Lucy G. Recchia<sup>1</sup> and Valerio Lucarini<sup>1</sup>

<sup>1</sup>University of Reading, Reading, Berkshire, RG6 6AH, U.K.

Correspondence: Lucy G. Recchia (l.g.recchia@reading.ac.uk)

**Abstract.** The South and East Asian summer monsoons are globally significant meteorological features, creating a strongly seasonal pattern of precipitation, with the majority of the annual precipitation falling between June and September. The stability of such a strongly seasonal hydrological cycle is of extreme importance for a vast range of ecosystems and for the livelihoods of a large share of the world's population.

Simulations are performed with an intermediate complexity climate model, PLASIM, in order to assess the future response of the South and East Asian monsoons to changing concentrations of aerosols and greenhouse gases. The radiative forcing associated with absorbing aerosol loading consists of a mid-tropospheric warming and a compensating surface cooling, which is applied to India, Southeast Asia and East China, both concurrently and independently. The primary effect of increased absorbing aerosol loading is a decrease in summer precipitation in the vicinity of the applied forcing, although the regional responses vary significantly. The decrease in precipitation is only partially ascribable to a decrease in the precipitable water, and instead derives from a reduction of the precipitation efficiency, due to changes in the stratification of the atmosphere.

When the When the absorbing aerosol loading is added in all regions simultaneously, precipitation in East China is most strongly affected, with a quite distinct transition to a low precipitation regime as the radiative forcing increases beyond 60 W/m². The response is less abrupt as we move westward, with precipitation in South India being least affected. By applying the absorbing aerosol loading to each region individually, we are able to explain the mechanism behind the lower sensitivity observed in India, and attribute it to aerosol forcing remote absorbing aerosol forcing applied over East China. Additionally, we note that the effect on precipitation is approximately linear with the forcing.

The impact of doubling carbon dioxide levels is to increase precipitation over the region, whilst simultaneously weakening the circulation. When the carbon dioxide and absorbing aerosol forcings are applied at the same time, the carbon dioxide forcing partially offsets the surface cooling and reduction in precipitation associated with the absorbing aerosol response. Assessing the relative contributions of greenhouse gases and aerosols is important for future climate scenarios, as changes in the concentrations of these species has the potential to impact monsoonal precipitation.

# 1 Introduction

35

The South and East Asian monsoon systems are of key importance for the regions region's economy, agriculture and industry. The monsoons bring the majority of the annual rainfall over the summer months. The seasonality of rainfall is a major controlling factor for agricultural activities as well as for natural ecosystems. Significant deviations from the normal onset dates and amount of summer rainfall associated with the monsoon has far-reaching consequences, especially for farmers (Krishna Kumar et al., 2004). Crop production, which contributes substantially to the regions Gross Domestic Product, is particularly adversely affected during deficit rainfall years (Gadgil and Gadgil, 2006; Zhang et al., 2017). The occurrence of flooding (Francis and Gadgil, 2006; Wang and Gu, 2016; Mishra, 2021) or droughts (Gadgil et al., 2002; Hazra et al., 2013; Francis and Gadgil, 2010; Krishnamurti et al., 2010; Zhang and Zhou, 2015), typically linked to strong or weak monsoon years, also have widespread socio-economic impacts and are expected to become more frequent in the future (Intergovermental Panel on Climate Change (IPCC), 2021).

There remains considerable uncertainty in how the monsoons will evolve in a changing climate, both in terms of the response of the monsoons to an applied forcing and in the nature of future climate forcing itself. Despite many improvements to state-of-the-art Global Climate Models (GCMs) (Choudhury et al., 2022; Khadka et al., 2022; Gusain et al., 2020; Xin et al., 2020; Jiang et al., 2020; Ogata et al., 2014; Sperber et al., 2013), difficulties persist in accurately portraying the physical processes and interactions associated with the South and East Asian monsoon systems (Ul Hasson et al., 2016). Variability between models remains an issue, with large inter-model spreads reducing the confidence in future climate projections. The multi-model mean usually performs better than any single model (Khadka et al., 2022; Gusain et al., 2020; Xin et al., 2020; Jiang et al., 2020). Systematic biases in GCMs typically arise from poor representations of orographic rainfall and the various oceanic phenomena affecting sea surface temperature (Wang et al., 2014; Annamalai et al., 2017; McKenna et al., 2020; Choudhury et al., 2022). Precipitation over northern India and the Yangtze River valley is underestimated, whilst precipitation on the leeward side of the Western Ghats and the mountainous eastern states is overestimated (Konda and Vissa, 2022; Gusain et al., 2020; Xin et al., 2020). Biases in precipitation are linked with biases in the moisture transport; models typically have too much moisture over the Indian peninsula and too little moisture over central India (Konda and Vissa, 2022) (Boschi and Lucarini, 2019; Konda and Vissa, 2022). The role of irrigation is largely ignored, despite its importance in determining the strength and spatial extent of local-scale precipitation (Ul Hasson et al., 2016) (Chou et al., 2018). It is also-widely accepted that a reasonable representation of the large-scale circulation is required for accurate simulation of precipitation (Xin et al., 2020; Khadka et al., 2022).

It is generally accepted In general, it is agreed upon that monsoonal precipitation in the northern hemisphere will increase, with more frequent extreme rainfall events, despite a weakening of the large-scale circulation (Intergovermental Panel on Climate Change (IPCC), 2021; Wang et al., 2021; Krishnan et al., 2020; Wester et al., 2019). In Changes to forcings, such as greenhouse gases and aerosols, can have significant impacts on local moisture availability and affect the hydrology. In combination with a strong extratropical event, such as prolonged atmospheric blocking or advection of a low-pressure system, this can lead to extreme floods and heat waves (Lau and Nath, 2012; Boschi and Lucarini, 2019; Galfi and Lucarini, 2021). In

the near-term future, internal variability is expected to remain the dominant mode in modulating the monsoonal precipitation, but by the end of the 21st century, the effects of external forcing will become prevalent. The main components of external forcing are greenhouse gases and aerosols, which impact the optical properties of the atmosphere over a variety of spectral ranges. Phenomena contributing to the internal variability on annual-decadal timescales include the Indian Ocean Dipole, the El Niño Southern Oscillation (ENSO), the Pacific Decadal Oscillation and the Atlantic Multidecadal Oscillation. ENSO is the most important contributor on interannual timescales and it is expected that the precipitation variability due to ENSO will increase (Intergovermental Panel on Climate Change (IPCC), 2021; Wang et al., 2021). Here, we focus on the South and East Asian monsoons response to external forcing, specifically to globally increased greenhouse gas concentration and regional absorbing aerosol loading.

In terms of the Asian monsoons, aerosols and greenhouse gases have competing effects. Greenhouse gases act to warm the surface, enhancing the contrast between land and sea temperatures and enabling greater moisture uptake (e.g. Douville et al. (2000); Ueda et al. (2000); Ueda et al. (2000); Held and Soden (2006); Cherchi et al. (2011); Samset et al. (2018); Cao et al. (2022)

). This leads to stronger monsoons and increased precipitation, despite a weakening of the large-scale circulation (Turner and Annamalai, 2012; Lau and Kim, 2017; Swaminathan et al., 2022; Intergovermental Panel on Climate Change (IPCC), 2021).

Generally, aerosols have a stabilising effect on the atmosphere, through surface coolingand mid-tropospheric warming, increas-

ing the stratification of the atmosphere and causing a drying trend which results in a weaker monsoon (Li et al., 2016; Wilcox et al., 2020; Ayantika et al., 2021; Cao et al., 2022). Historically, aerosol forcing has dominated, linked with a declining rainfall trend over the latter half of 20th century (Bollasina et al., 2011; Polson et al., 2014; Li et al., 2015; Undorf et al., 2018; Dong et al., 2019)

(Bollasina et al., 2011; Polson et al., 2014; Li et al., 2015; Undorf et al., 2018; Dong et al., 2019; Ramarao et al., 2023), but moving forward, greenhouse gas forcing is expected to dominate, which is associated with a likely increase in monsoonal rainfall in the northern hemisphere (Monerie et al., 2022). In the near-term, changes in aerosol concentrations over the Asian region will continue to impact the monsoons, with increases in anthropogenic emissions acting to weaken the large-scale circulation and hydrological cycle, thus weakening the monsoons, whilst decreases in emissions, perhaps through the enforcement of air quality policies, will likely intensify the monsoons.

Aerosols are often referred to as scattering or absorbing, depending on how the aerosol interacts with shortwave radiation. Scattering aerosols, such as sulphates, cause surface cooling through scattering radiation, which weakens the land-sea temperature gradient. This inhibits moisture advection from the Arabian Sea and reduces low-level southwesterly wind speeds, resulting in decreased precipitation (Sherman et al., 2021). Absorbing aerosols, the most important of which is black carbon, cause both surface cooling and localised mid-level heating by absorbing some of the incoming radiation. The effect of black carbon on the South and East Asian monsoons is debated. Meehl et al. (2008) found that whilst black carbon contributed to increased precipitation during the pre-monsoon period, there was a decrease in monsoonal precipitation. In partial agreement, Lau and Kim (2006) also find a link between increased black carbon (and dust) loading and pre-monsoon precipitation, but that subsequent precipitation decreased over India, and increased over East Asia. Kovilakam and Mahajan (2016) noted a near-linear relationship between black carbon loading and summer precipitation in the South Asian monsoon, with enhanced monsoonal circulation, stronger meridional tropospheric gradient and increased precipitable water as the levels of black carbon

are raised. Others have found that black carbon weakens the Asian monsoons (Chakraborty et al., 2004; Liu et al., 2009) or that the results are inconclusive (Sanap and Pandithurai, 2015; Sherman et al., 2021). Although Guo et al. (2016) suggest that sulphates are the dominant aerosol species for impacting South Asian monsoon rainfall, further work is needed to unify the individual effects of different aerosol species on the vertical profile of the atmosphere as well the interactions between them (Lu et al., 2020).

The COVID-19 pandemic in 2020 offered an unprecedented opportunity to observe the effects of reduced aerosol loading, resulting from nationwide lockdowns, on the South and East Asian monsoons. The decrease in aerosols was found to correspond to a 4 W/m² increase in surface solar radiation (Fadnavis et al., 2021), resulting in warmer surface temperatures over land. The heightened contrast in land-sea temperature and stronger low-level winds enhanced moisture inflow to the monsoon system, leading to increased precipitation and a more intense monsoon (Fadnavis et al., 2021; Lee et al., 2021; Kripalani et al., 2022; He et al., 2022). In particular, spatial coherence between reduced aerosol loading and precipitation has been noted over West-Central India and the Yangtze River Valley (Kripalani et al., 2022).






The South Asian summer monsoon has been identified as a possible tipping point (Lenton et al., 2008), highlighting the importance of investigating the impact of changing radiative forcing. Radiation plays a major role in the evolution of the monsoon, with seasonal changes in temperature and circulation triggering monsoon onset. The onset itself can be considered a regime change, with a simplified model able to reproduce the transition from low to high relative humidity as the low-level wind speed is increased (Recchia et al., 2021). Modelling of the Indian monsoon by Zickfeld et al. (2005) suggests that an abrupt transition to a low rainfall regime with a destabilised circulation is plausible, given changes to the planetary albedo, the insolation and/or the carbon dioxide concentration. The possibility of an abrupt transition (on intraseasonal timescales) is supported by Levermann et al. (2009), who use a similar conceptual model based on heat and moisture conservation to show the bistability of the monsoon system, given net radiative influx above a critical threshold. Schewe et al. (2012) take this conceptual model a step further, so that it can be applied on orbital timescales, and demonstrate a series of abrupt transitions of the East Asian monsoon that corresponds to a proxy record of the penultimate glacial period. In contrast, Boos and Storelymo (2016) use a simplified energetic theory in combination with GCM simulations to show that the intensity of summer monsoons in the tropics has a near-linear dependence on a range of radiative forcings. They argue that previous tipping point theory neglected the static stability of the troposphere and that no threshold response to large radiative forcings is observed in a GCM simulation, on decadal timescales. Kovilakam and Mahajan (2016) also find no abrupt transitions of the South Asian monsoon in GCM simulations with very high black carbon loading, despite significant surface cooling and a decrease in global precipitation.

The main goal is to understand the impact of <u>absorbing</u> aerosol forcing on the monsoon in India, Southeast Asia and East China, focusing on the precipitation and the large-scale circulation, using a simplified modelling framework. We will investigate the level of radiative forcing required for a breakdown of the monsoon circulation and capture the transition from a high to a low precipitation regime. Additionally, we wish to explore the combined effect of regional <u>absorbing</u> aerosol forcing with globally increased carbon dioxide concentration, on the South and East Asian monsoons. The <u>so-called Silk road pattern</u> indicates the presence of westward propagating signal linking the Indian monsoon with atmospheric patterns over China as

a result of Rossby waves (Enomoto et al., 2003; Ding and Wang, 2005; Chakraborty et al., 2014; Boers et al., 2019). We will also consider whether localised forcings applied in each of the regions of interest have noticeable remote impacts.

We use an intermediate complexity climate model, the Planet Simulator (PLASIM) (Fraedrich et al., 2005; Lunkeit et al., 2011), to simulate the response of the South and East Asian monsoons to forcing scenarios with differing levels of carbon dioxide and absorbing aerosols. Details of the experiment set-up and design are given in Section 2. There are several studies using idealised models based on conservation of heat and moisture (Zickfeld et al., 2005; Levermann et al., 2009; Schewe et al., 2012; Boos and Storelvmo, 2016) to investigate the occurrence of regime transitions in the monsoons, and a multitude of literature using complex GCMs to simulate future climate response to different forcings scenarios (Katzenberger et al., 2021; Almazroui et al., 2020; Varghese et al., 2020; Wang et al., 2016; Ogata et al., 2014; Song et al., 2014), but little using intermediate complexity models (Wang et al., 2016; Herbert et al., 2022). Intermediate complexity models are able to reproduce large-scale climate features with a fair degree of accuracy, whilst maintaining the flexibility to perform long simulations at low computational cost and to allow for parametric exploration. However, they are unable to capture small scale convection or oceanic processes such as ENSO. We aim to help bridge the gap between conceptual and complex models.

The results of the simulations are presented in Sections 3 and 4, with the former focusing on the response of key variables such as precipitation and circulation to absorbing aerosol forcing, and the latter considering the combined effect of carbon dioxide and absorbing aerosol forcing on the South and East Asian monsoons. Additionally, we consider the relationship between the added forcing and regionally-averaged summer precipitation, trying to capture the signature of abrupt transitions. In Section 5 we show the results of applying absorbing aerosol forcing to individually to regions of India, Southeast Asia and East China, rather than simultaneously (as in Sections 3 & 4), which allows attribution of responses to local and remote forcing, within the Asian region. Furthermore, we can compare the linear combination of response due to local-scale absorbing aerosol forcing to the response from regional-scale forcing, in the spirit of response theory (Ragone et al., 2016; Lucarini et al., 2017) linear response theory (Ragone et al., 2016; Lucarini et al., 2017; Ghil and Lucarini, 2020).

# 2 Data & Methods





The PLASIM model (Lunkeit et al., 2011; Fraedrich et al., 2005) is an intermediate complexity global climate model, with a dynamical core and parameterised physical processes. It includes parameterisation schemes for land-surface interactions, radiation and convection. There are several versions of the PLASIM model, where adaptions have been made to allow inclusion of additional climatic components such as vegetation, sea ice and ocean (Holden et al., 2016; Platov et al., 2017). We use the version developed by von Hardenberg (2020), which features a large-scale geostrophic ocean component. Vertically, there are 10 atmospheric levels and horizontally, the spectral resolution is T42, which approximately corresponds to 2.8°. When focusing on is on large-scale features over long timescales, the PLASIM model is an invaluable tool for investigating various future scenarios, at a low computational cost.

The PLASIM model has previously been used to investigate regime transitions on a range of timescales, including the arid to hyperarid transition of the Atacama Desert (Garreaud et al., 2010) and the effect of a permanent El Niño state on susceptible

tipping elements like the Amazon rainforest and the Sahel Belt (Duque-Villegas et al., 2019). PLASIM has also been used to simulate global-scale transitions between a warm (current climate) and a snowball state (Boschi et al., 2013; Lucarini et al., 2013), which has helped to advance understanding of the multistability properties of the Earth's climate system (Lucarini et al., 2010; Gómez-Leal et al., 2018; Margazoglou et al., 2021). In terms of future climate projections, the PLASIM model has been used as an emulator (Holden et al., 2014; Tran et al., 2016) and to support the application of Ruelle's response theory (Ragone et al., 2016; Lucarini et al., 2017).

There are multiple precedents for using both PLASIM and other models of a similar complexity for climate simulations regarding the Asian monsoons (Wang et al., 2016; Thomson et al., 2021; Herbert et al., 2022). In terms of lower complexity models, Zickfeld et al. (2005) use a box model of the tropical atmosphere to investigate the stability of the South Asian monsoon to changes in planetary albedo and carbon dioxide concentration. For higher complexity models, there is an abundance of literature where CMIP5 and CMIP6 standard models are used to explore the response of the South and East Asian monsoons to future climate scenarios (Menon et al., 2013; Li et al., 2015; Kitoh, 2017; Swapna et al., 2018; Varghese et al., 2020; Krishnan et al., 2020; Almazroui et al., 2020; Chen et al., 2020; Moon and Ha, 2020; Wang et al., 2020, 2021; Swaminathan et al., 2022; Intergovermental Panel on Climate Change (IPCC), 2013, 2021). Here, we use a combination of our own simulations with the PLASIM modeland results from existing literature that use a hierarchy of models. We use the intermediate complexity model, PLASIM, to quantify the responses of the South and East Asian monsoons to a range of future climate scenarios, thereby contributing to the existing literature and ensuring that a hierarchy of models are represented.

# 2.1 Model validation







A brief evaluation of the PLASIM model is conducted, in order to show that the PLASIM model is capable of reproducing the key features of the South and East Asian monsoons to a sufficient degree of accuracy for our purposes. Similar versions of the PLASIM model have been shown to perform well in climate simulations (Holden et al., 2016; Platov et al., 2017). Figures 1–4 show the performance of a 50-year 100-year control simulation with the PLASIM model against ERA5 reanalysis data averaged over years 1988–2017 (Copernicus Climate Change Service, 2017) for key variables of surface temperature, relative humidity, circulation and precipitation. This The PLASIM control simulation has no absorbing aerosol forcing and uses present-day carbon dioxide concentration of 360 ppmv. The left column shows the June-July-August (JJA) average, and the right column the December-January-February (DJF) average.

There is a clear difference between summer and winter (JJA) and winter (DJF) in Figures 1–4, with the PLASIM model capturing the strong seasonality in the Asian region and in fair agreement with ERA5 reanalysis data. Figure 1 shows the increase in surface temperature of the Eurasian landmass in summer, creating a thermal contrast with the relatively cooler ocean. North of the Himalayas, the accelerated heating of the Tibetan Plateau can be seen, which plays a pivotal role in the development of the South and East Asian monsoons.

The seasonal reversal in low-level winds is distinctly visible in Figure 2. In winter, the 850 hPa winds over South Asia are easterly, but in summer, they are westerly. PLASIM overestimates the dryness to the north and west of India in summer, but underestimates the westerly winds over South Asia, compared to ERA5 reanalysis data. During winter, PLASIM has higher

Figure 1. Surface Land & sea surface temperature (shading) and terrain height (grey-white/black contours) from 50-year PLASIM control run, averaged over June-July-August (left column) and December-January-February (right column). Top row shows data from 100-year PLASIM control run. Bottom row shows data from ERA5 reanalysis (Copernicus Climate Change Service, 2017) for period 1988–2017.

easterly wind speeds at 10-15°N for the 850 hPa level than ERA5 reanalysis data. At 200 hPa (Figure 3), the Tropical Easterly Jet is present over the summer months. The formation of the Tropical Easterly Jet is linked with the area of high pressure over the Tibetan Plateau and the resulting anticyclone. Figure 3 also shows the migration of the westerly subtropical jet from south of the Tibetan Plateau in winter, to north of the Tibetan Plateau in summer. The subtropical jet is shown with lower wind speeds in summer for PLASIM than ERA5 reanalysis, although the wind speeds are more comparable between the datasets in winter. The relative humidity at high levels over the Asia continent is lower in ERA5 reanalysis than PLASIM, particularly during winter.




Warmer temperatures in summer are associated with drier air inland, and more humid conditions in coastal regions. The subtropical jet brings dry air at mid-high levels towards East Asia (Figure 3). In summer, a convergence line forms along the India-Pakistan border where the dry air mass from the Eurasian continent meets the moist monsoonal air mass. There is a significant contrast in low-level relative humidity over South and East Asia between summer and winter (Figure 2), and an even greater contrast in the precipitation (Figure 4). The majority of South Asia receives only a few millimeters of rain per day in winter, whereas at least triple the amount falls in summer, and even more in the mountainous areas.

**Figure 2.** Relative humidity (shading) and wind speed & direction (vectors) from 50-year PLASIM control run, at the 850 hPa level, averaged over June-July-August (left column) and December-January-February (right column). Areas of high orography are masked in grey Top row shows data from 100-year PLASIM control run. Bottom row shows data from ERA5 reanalysis (Copernicus Climate Change Service, 2017) for period 1988–2017. Data has been extrapolated below ground.

Despite the PLASIM model's low resolution and simplified physics, the precipitation over land is in reasonable agreement with ERA5 reanalysis data. PLASIM underestimates the amount of summer rainfall along the west coast of India and over central India, compared to ERA5 reanalysis, but captures the regions of highest precipitation - Bhutan & eastern states of India - well. The large-scale features of the monsoons, including the characteristically strong seasonality, is also well represented.

# 210 2.2 Experiment design


To analyse the roles of absorbing aerosols and greenhouse gases, which have contrasting effects, on the South and East Asian monsoons, we implement two forcing scenarios with the PLASIM model: *aerosol only* and *aerosol with 2xCO*<sub>2</sub>. The *aerosol only* simulation features increasing absorbing aerosol forcing, whilst the *aerosol with 2xCO*<sub>2</sub> simulation features doubled carbon dioxide levels (720 ppmv) to represent higher levels of greenhouse gases, as well as increasing absorbing aerosol forcing. This is a substantially simplified version of IPCC 6 forcing scenario SSP3-7.0 (Intergovermental Panel on Climate Change (IPCC), 2021).

**Figure 3.** Relative humidity (shading) and wind speed & direction (vectors) from 50-year PLASIM control run, at the 200 hPa level, averaged over June-July-August (left column) and December-January-February (right column). Areas of high orography are masked in grey Top row shows data from 100-year PLASIM control run. Bottom row shows data from ERA5 reanalysis (Copernicus Climate Change Service, 2017) for period 1988–2017. Data has been extrapolated below ground.

The PLASIM model has no explicit treatment of aerosol interactions, so we need to resort to a simplified representation of their effects on the radiation budget of the atmosphere. Along the lines of Chakraborty et al. (2004), we emulate the effect of absorbing aerosol loading by applying a varying mid-level tropospheric heating, say H, over three vertical levels, approximately corresponding to 550–750 hPa. The Thus, each of the three vertical levels has an applied forcing of intensity H/3. The absorbing aerosol forcing is applied simultaneously over three regions: India, Southeast Asia and East China, as per Figure 5. We also consider the impact of forcing in each of these regions separately (see Section 5). To the first approximation, the effect of absorbing aerosols is to redistribute the absorption of solar radiation in the atmospheric column and at the surface (e.g. Sarangi et al. (2018)). Hence, the surface is cooled by applying a compensating forcing of the same intensity as the mid-tropospheric heating, -H, which accounts for less solar radiation reaching the surface.



All simulations begin from steady state conditions, which have been obtained by discarding the transient portion of the runs. For both simulations, *aerosol only* and *aerosol with*  $2xCO_2$ , the <u>absorbing</u> aerosol forcing is gradually increased from  $0W/m^2$  to  $150W/m^2$ , and then decreased back to  $0W/m^2$ . In order to make sure that the system remains sufficiently close to steady state

**Figure 4.** Precipitation from 50-year PLASIM control run, averaged over June-July-August (left column) and December-January-February (right column). Top row shows data from 100-year PLASIM control run. Bottom row shows data from ERA5 reanalysis (Copernicus Climate Change Service, 2017) for period 1988–2017.

at all times, this takes place over a simulation length of 900 years. We consider 30W/m² to be low forcing, 60W/m² medium forcing and 90W/m² high forcing. These values are within observed ranges of radiative forcings due to aerosol loading of the atmosphere (Kumar and Devara, 2012; Vaishya et al., 2018). Aerosol Absorbing aerosol forcing above 100W/m² is deemed unrealistic in real-world terms, but such forcing values are considered in order to cover a parametric range of forcings, which includes the possibility of a breakdown or substantial weakening of the monsoon systems. The absorbing aerosol forcing does not follow seasonal modulation, in contrast to real-world scenarios—and other studies (e.g. Herbert et al. (2022)). We anticipate that there will be no noticeable hysteresis in the model's behaviour as a result of the of applied forcing.

# 3 Response to absorbing aerosol forcing



The added <u>absorbing</u> aerosol forcing causes the surface to cool in the regions where the forcing is applied, whilst also causing a warm anomaly around 700 hPa. The increased stratification of the atmosphere and the reduction of the land-sea temperature

Figure 5. Regions showing where absorbing aerosol forcing has been applied. Shading indicates terrain height (m).





contrast suppresses precipitation and weakens the large-scale circulation. As the <u>absorbing</u> aerosol forcing is increased, the response of the monsoons becomes more pronounced.

Figures 6–78 (and Figures S1–S3S6) are presented as panels of 2 x 6 panels rows by 3 columns for each variable, with the top left panel showing the state of the system at approximately  $30\text{W/m}^2$  of heating. The remaining five panels show the anomaly with respect to the control simulation (*aerosol only* - control) at forcings of approximately 30, 60, 90, 120 and  $150\text{W/m}^2$ . With the exception of  $150\text{W/m}^2$ , each panel in the figures is produced using the average of the two 10-year means centred around the respective forcing values. The panel for  $150\text{W/m}^2$  represents the single 10-year period centred on  $150\text{W/m}^2$ . Since there is no significant hysteresis in the simulations, the average over both the ascending branch with forcing  $0 \rightarrow 150\text{W/m}^2$  and the descending branch with forcing  $150 \rightarrow 0\text{W/m}^2$  is taken. In what follows Stippling is added to show statistically significant changes, defined as where the anomaly (*aerosol only* - control) is greater than double the JJA interannual variability of the *aerosol only* simulation. In the following text, we discuss the response of the system by looking at the mean summer (June, July, and August – JJA) precipitation, evaporation, surface temperature, and winds at 850 hPa and 200 hPa. Areas of high orography will be masked in grey for certain pressure levels.

The application of radiative forcing, associated with the presence of absorbing aerosols, leads to surface cooling and midlevel warming where anomalous absorption is present, as expected (Figure S1). When the intensity of absorbing aerosol forcing reaches approximately 90W/m², some areas of India and South East Asia have cooled by 10°C. Above 90W/m², the surface temperature drops more rapidly and becomes unrealistically cool at -15°C in parts of East China when the forcing is close to 150W/m². The intense surface cooling is primarily responsible for activating the ice-albedo feedback when temperatures are close the freezing, and the effect effect; a positive feedback which enhances surface cooling as sea ice and snow cover increases, causing a greater amount of radiation to be reflected back into space. The result is similar to the establishment of a nuclear winter, albeit via a slightly different mechanism. In our modelling strategy, the aerosol vertically redistributes effect of absorbing aerosols is emulated by vertically redistributing the heating, rather than removing it from the atmospheric column.

**Figure 6.** Aerosol only simulation. Contours showing mean decadal June-July-August precipitation (first column, top row) & precipitable water (first column, third row), and mean decadal June-July-August precipitation anomaly (top two rows), and mean decadal June-July-August precipitable water & precipitable water anomaly (bottom two rows), compared to the control run (anomaly = aerosol only - control), for a range of absorbing aerosol forcing values. Stippling where the anomaly exceeds double the JJA interannual variability.

The extreme cold surface temperature indicates that our results lose physical meaning for values of forcing intensity greater than  $\sim 100 \text{W/m}^2$ . At the 700 hPa level (Figure S1, bottom two rows), a strong statistically significant warm anomaly develops over East China, which becomes stronger as the forcing increases. There is also a slight warming at little change in 700 hPa

**Figure 7.** Aerosol only simulation. Contours showing mean decadal June-July-August wind speed & direction (shading & vectors) specific humidity and mean decadal June-July-August wind speed & direction specific humidity anomaly (shading & vectors) compared to the control run (anomaly = aerosol only - control), for a range of aerosol forcing values. The top two rows are at 850-925 hPa and the bottom two rows at 200-700 hPa. Areas of high orography are masked in grey. Stippling where the anomaly exceeds double the JJA interannual variability.

over North India. Both of these areas with anomalous warming at temperature across North India and Myanmar, with only
the southern peninsulas of India and Southeast Asia showing cool anomalies at high to extreme levels of forcing. The areas showing little cooling or warming at the 700 hPa level, namely, northwest India, East India-Myanmar and East China, are also

**Figure 8.** *Aerosol only* simulation. Contours showing mean decadal June-July-August wind speed & direction (shading & vectors) and mean decadal June-July-August wind speed anomaly (shading) compared to the control run (anomaly = *aerosol only* - control), for a range of absorbing aerosol forcing values. The top two rows are at 850 hPa and the bottom two rows at 200 hPa. Areas of high orography are masked in grey. Stippling where the anomaly exceeds double the JJA interannual variability.

associated with anomalous high pressure (Figure S3). Despite the same forcing being applied over Southeast Asia, there is no corresponding warming at the 700 hPa levelS2). The monsoon trough, a low pressure region to the north of India that is associated with the formation of the Indian monsoon, intensifies as the forcing increases beyond 90W/m². When the absorbing

aerosol forcing is greater than 90W/m<sup>2</sup>, a cold anomaly forms statistically significant cold anomaly develops over the Middle East, creating an East-West temperature dipole. In general, the combination of surface cooling and mid-level warming, clearly seen in vertical cross-sections of temperature (Figure S4), leads to a strong temperature inversion and an increase in the static stability of the atmosphere, which suppresses moist convective processes and thus reduces precipitation.







Figure 6 (top two rows) illustrates the reduction in precipitation, corresponding with increased radiative forcing. The East China and Southeast Asia regions are the most affected. With the exception of the west coast and northeastern states, areas of higher orography, the majority of India does not follow the same trend of declining rainfall. Partly, this is due to the low-level wind, which brings a large influx of moisture from over the Arabian Sea (discussed further below). In contrastAt high to extreme levels of forcing, eastern Siberia experiences a reduction in precipitation, despite being outside the area where forcing is applied.

In order to better understand the reasons behind the decrease To understand the variation in the precipitation, we look into the vertically integrated precipitable water, response to absorbing aerosol forcing, other variables such as the vertically integrated precipitable water, specific humidity, evaporation and circulation are considered.

Figure 6 (bottom two rows), showing the change in precipitable water with absorbing aerosol forcing, highlights a moist anomaly over the Middle East, and to a lesser extent, the Indian Ocean. Advection of moisture from East to West helps explain why precipitation over India is not reduced as much as over dry anomalies over northeast India, Southeast Asia and East China. It is notable that the precipitable water does not decrease at as fast a rate as the rainfall decreases. Significant reductions in the precipitable water only occur at a high intensity of forcing, specifically over These anomalies only become statistically significant at 90W/m<sup>2</sup>. Therefore, the decline in rainfall forcing and above, in contrast to the changes in the precipitation which become significant from 60W/m<sup>2</sup>. The fact that the precipitation decreases at a faster rate than the precipitable water is primarily attributed to a reduction in precipitation efficiency, which is due to the increase in the static stability of the lower levels of the atmosphere, as opposed to a searcity of moisture availability reduction in the moisture availability of the atmospheric column.

There is a slight moist anomaly over India up to 60W/m<sup>2</sup> forcing. Beyond 60W/m<sup>2</sup> forcing, there remains a region in central India that doesn't experience the significant reduction in precipitable water seen over Southeast Asia and East China. Similarly, looking at Figure 7, the specific humidity at 925 hPa is higher over north India up to 90W/m<sup>2</sup> forcing, and thereafter the specific humidity reduces significantly in the area of applied forcing, except for a small region in west-central India. A vertical cross-section of specific humidity along 20°N (Figure S5, top two rows) also highlights the moist anomaly over North India (70–90°E) at 30 and 60W/m<sup>2</sup>, and the moist anomaly over the Middle East (around 50°E).

Looking at another element of the hydrological cycle, namely the evapotranspiration, can provide additional insight (Figure S2From Figures 7 and S5, the greatest decline in specific humidity occurs near the surface, which corresponds with a reduction in the evaporation (Figure S3). The absorbing aerosol forcing leads to a decrease in the evaporation because of the reduction of the solar radiation reaching the surface, following a similar spatial pattern to the precipitation. However, the decline in evaporation is less pronounced than the decline in precipitation, further supporting the idea that the reduction in rainfall is not solely due to a lack of moisture in the atmosphere. The contrasting response of the evaporation and precipitation over

India indicates that the anomalous behaviour of the latter is likely to be related to non-local effects, which is explored below in Section 5 (Ramarao et al., 2023). There is a greater decrease in evaporation over east-central and southern India, than over northwest India, in agreement with observed evapotranspiration trends in the period 1979–2008 Ramarao et al. (2023).

Another key effect of the applied radiative forcing is to weaken the large-scale circulation. At low levels (Figure 78, top two rows), there is a reduction in strength of the southwesterly wind in the band 0–20°N, which is a key driver of both the South and East Asian monsoons. With approximately 60W/m<sup>2</sup> of heating, the wind speed is reduced by 2-3 m/s, and with 90W/m<sup>2</sup> heating, there is a 4-5 m/s reduction in wind speed. When the maximum forcing is applied, the speed of the southwesterly monsoon wind becomes close to zero. Stronger radiative forcing causes more pronounced surface cooling, weakening the land-sea temperature gradient and thus weakening the low-level monsoon flow.






There is a strengthening of the low-level southwesterly wind in East China, causing (20-40°N), causing more dry air to be advected towards East Siberia from southwest to northeast, towards Eastern Siberia, and corresponding to a reduction of precipitation in the region. The mid-level wind field (not shown) experiences changes of a similar magnitude to the low-level wind field. A narrow region stretching from northwest India to Bangladesh, and continuing northeastwards, features increased wind speeds, which is associated with the formation of three atmospheric highs (Figure \$3\$S2).

At higher levels (Figure 78, bottom two rows), there is a significant reduction in the speed of the Tropical Easterly Jet, from Southeast Asia to the east coast of Somalia. Higher absorbing aerosol forcing corresponds with greater declines in easterly wind speeds. Additionally, there is a slight weakening of the westerly subtropical jet, located north of India, at high forcing rates (>90W/m<sup>2</sup>).

With light ( $30\text{W/m}^2$ ) absorbing aerosol forcing, there is vertical ascent over land for the regions South India, Southeast Asia and East China (Figure S6), corresponding with the locations of peak convective rainfall (Figure S6 - dotted lines). Note that Figure S6 shows the vertical velocity in pressure coordinates ( $\omega$ ), so that negative vertical velocity corresponds to ascent, and positive vertical velocity to descent. As the absorbing aerosol forcing increases, the upwards vertical velocity reduces, so that there is little vertical motion. This is consistent with increased atmospheric stratification and increased static stability, leading to suppression of convective precipitation.

Precipitation (top), precipitable water (middle) and surface temperature (bottom), averaged over the regions indicated (following the marked boxes in Figure 5), for *aerosol only* and *aerosol with* 2xCO<sub>2</sub> runs, against the aerosol forcing. Precipitation is separated into convective and large-scale components. Variables taken as a running 20-year June-July-August mean. From Figure S6 (top two rows), there is an area of vertical ascent around 20°N, 70–80°E, which becomes stronger as the absorbing aerosol forcing increases. Although there is not an equivalent increase in the convective precipitation, from Figure 6, there is a slight increase in total precipitation for central India at high-extreme levels of forcing. Thus, the anomalous precipitation over central India comes from large-scale rather than convective processes, and is linked to changes in the circulation, likely related to non-local effects (explored further in Section 5).

# 3.1 Relationship between forcing and precipitation




Here we consider the behaviour of the precipitation, averaged over key regions of North India, South India, East China and Southeast Asia, as the radiative forcing increases. India is separated into north and south regions, as the responses are significantly different in the two sub-regions. The precipitation is split into large-scale and convective components. The regionally-averaged vertically-integrated precipitable water are regionally-averaged surface temperature are also investigated, to understand if changes in precipitation correlate with changes in the precipitable water or surface temperature. Note that the area-averages refer to land points only.

Looking firstly at the precipitation (Figure &9, top), we can see that the convective component dominates. As the forcing increases, the convective precipitation reduces; however, the large-scale precipitation slightly increases. There is significant variation in convective precipitation response between the regions, with East China showing the most abrupt decline and South India being the least impacted. The response of South India (grey solid line) to the applied radiative forcing, which causes a cooling and drying effect, is partially mitigated by advection of moisture from the surrounding oceans. For North India (black solid line), there is a nearly linear decline in convective precipitation as the forcing increases. As noted above, East China (red solid line) experiences an abrupt decline at approximately 60W/m² forcing, after which the convective precipitation drops to a negligible amount. The behaviour of the convective precipitation in Southeast Asia (blue solid line) is somewhere between that of East China and North India: it decreases more sharply than North India, but without the abrupt transition at around 60W/m². The large scale precipitation simulated by the model is only weakly affected by the absorbing aerosol forcing, and its response partially compensates the severe decrease of the convective precipitation.

**Figure 9.** Precipitation (top), precipitable water (middle) and surface temperature (bottom), averaged over the regions indicated (following the marked boxes in Figure 5), for *aerosol only* and *aerosol with*  $2xCO_2$  runs, against the absorbing aerosol forcing. Precipitation is separated into convective and large-scale components. Variables taken as a running 20-year June-July-August mean.

The regional variation of area-averaged precipitable water (Figure 89, middle) is less than that of precipitation. There is a transition in the precipitable water for all regions, from approximately constant to linearly declining as the forcing increases past 60W/m<sup>2</sup>. We note again that between 0 and 60W/m<sup>2</sup> forcing, although the convective precipitation is gradually reducing, the precipitable water does not correspondingly decline. This is attributed to a reduction in the precipitation efficiency, associated with changes in the dynamics of the atmosphere.

The area-averaged surface temperature (Figure §9, bottom) initially follows the same pattern as precipitable water, remaining roughly constant as the radiative forcing increases from 0 to 60W/m². At low (<60W/m²) absorbing aerosol forcing, the suppression of evaporative and convective processes appears to offset the surface cooling associated with the presence of absorbing aerosols. After 60W/m², the surface temperature declines with increasing forcing, at a near-linear rate for North India and Southeast Asia. For East China and South India, the rate of surface cooling increases at around 120-130W/m², which is linked with triggering ice-albedo feedback. For the range of absorbing aerosol forcing 0–80W/m², the surface temperature is only weakly sensitive to the forcing, whilst the precipitation is highly sensitive. Beyond 80W/m², the sensitivity reverses, with surface temperature being strongly affected by increasing forcing.

**Table 1.** Slopes for Figure 9, taken as a linear fit for the ranges 0–60 and 60–80 W/m<sup>2</sup> absorbing aerosol forcing, for *aerosol only* simulation.

|                      | Convective precipitation (mm/day per W/m²) |        | Č                       | le precipitation<br>y per W/m <sup>2</sup> ) | 1      | able water<br>per W/m <sup>2</sup> ) | Surface temperature (°C per W/m²) |        |  |
|----------------------|--------------------------------------------|--------|-------------------------|----------------------------------------------|--------|--------------------------------------|-----------------------------------|--------|--|
| Forcing range (W/m²) | 0-60                                       | 60-80  | 0-60                    | 60-80                                        | 0-60   | 60-80                                | 0-60                              | 60-80  |  |
| North India          | -0.019                                     | -0.054 | 0.005                   | 0.008                                        | 0.008  | -0.078                               | -0.060                            | -0.069 |  |
| South India          | -0.001                                     | -0.008 | 0.009                   | 0.007                                        | 0.009  | -0.027                               | -0.032                            | -0.042 |  |
| East China           | -0.058                                     | -0.115 | $\underbrace{0.008}_{}$ | $\underbrace{0.008}_{\bullet}$               | 0.005  | -0.154                               | -0.024                            | -0.103 |  |
| Southeast Asia       | -0.068                                     | -0.063 | 0.006                   | 0.023                                        | -0.030 | -0.086                               | -0.017                            | -0.117 |  |

#### 3.2 Summary & discussion


In summary, applying a regional absorbing aerosol forcing consisting of a mid-tropospheric heating and an equivalent surface cooling results in suppression of precipitation and a weakening of the large-scale circulation in the region. Areas of high pressure form, most prominently over East China. Some effects, such as the reduction in precipitation, extend to eastern Siberia. There is a much greater decline in precipitation over Southeast Asia and East China, than over India. The weakening of both the Indian South and East Asian monsoons in response to absorbing aerosol forcing has been observed and modelled (e.g. Lau and Kim (2010); Bollasina et al. (2011); Ganguly et al. (2012); Song et al. (2014); Dong et al. (2019)). Our simulation results are in agreement with the results of CMIP5 and CMIP6 standard models (Song et al., 2014), in particular with Ayantika et al. (2021) and their historic simulations with the IITM Earth System Model (version 2).

A more quantitative angle can be taken on the sensitivity of the monsoon characteristics with respect to the intensity of the absorbing aerosol forcing by considering Figure 89 and Table 1, focusing on the area-averaged precipitation, area-averaged precipitation area-averaged surface temperature. As the radiative forcing is increased form from 0 to 150W/m², there is a monotonic decrease in the convective precipitation for the regions of North India and Southeast Asia, with the decrease being slightly more pronounced for Southeast Asia than for North India. East China shows the highest sensitivity to aerosol forcing of approximately 60Wabsorbing aerosol forcing, and experiences the greatest rate change of convective precipitation from -0.058 at 0-60W/m², where to -0.115 at 60-80W/m² (from Table 1). After 80W/m², the convective precipitation drops to almost zero; essentially a breakdown of the monsoon system.

In contrast to the convective precipitation, the amount of precipitable water is only weakly affected by the <u>absorbing</u> aerosol forcing in the range 0–60W/m², as indicated by the small slope values in Table 1, and thereafter monotonically decreases with larger forcing. The surface temperature is also only weakly affected by <del>aerosol forcing up to around 80W low-medium intensity absorbing aerosol forcing.</del> There is little change in the rate of surface temperature between 0–60W/m² and 60–80W/m² for North and South India, but a greater rate of decline is noted for East China and Southeast Asia (Table 1). Beyond 80W/m², the surface temperature declines rapidly with increasing forcing, with evidence of the ice-albedo feedback activating in East China and South India at 120–130W/m². This analysis further confirms that the dramatic reduction in the precipitation is not due to a drying of the atmospheric column.

# 4 Response to combined absorbing aerosol and greenhouse gas forcing







Enhanced carbon dioxide levels lead to higher surface temperatures, higher absolute humidity levels and weakening of the large-scale circulation (Held and Soden, 2006), as discussed in Section 1. Here, we investigate the response of the South and East Asian monsoons to combined absorbing aerosol and carbon dioxide forcing, and whether doubling the carbon dioxide levels is sufficient to offset the aerosol effects of surface cooling and reduced precipitation.

Doubling the carbon dioxide levels leads to warmer surface temperatures, with regions of South Asia being several degrees warmer in the *aerosol with*  $2xCO_2$  simulation than in the *aerosol only* simulation (Figure \$4\$7, top row); however, for absorbing aerosol forcing over  $60\text{W/m}^2$ , the net effect of combined absorbing aerosol and carbon dioxide forcing remains—is a surface cooling, for aerosol forcing over  $60\text{W/m}^2$ . At the 700 hPa level, (Figure \$7, bottom row), the higher level of carbon dioxide also leads to several degrees of warming, compared to absorbing aerosol forcing alone (*aerosol only* simulation). From Figure \$9, the warming from enhanced carbon dioxide levels is greater at mid-levels than at the surface, relative to the *aerosol only* simulation.

Given the warmer surface temperatures in the *aerosol with*  $2xCO_2$  simulation, we would expect to see greater atmospheric moisture content and higher rates of evaporation and precipitation, compared to the *aerosol only* simulation. Although Both the precipitable water is considerably and specific humidity are significantly greater when carbon dioxide levels are doubled (Figure 910, bottom row), there is only a slight increase in the rates. Figures S8 & S10). The change in specific humidity is most pronounced close to the surface (Figures S8 & S10) and over the Middle East (around 20°N, 50°E), enhancing the moist

anomaly there. In contrast, there is no significant change in the rate of evaporation (not shown) and precipitation (Figure 9, top row), indicating that the aerosol forcing is dominating over land for the *aerosol with 2xCO*<sub>2</sub> simulation compared to *aerosol only*. There is an increase in both the evaporation (significant) and the precipitation (moderate) over the Indian Ocean when carbon dioxide levels are doubled.


In terms of the precipitation, there are some differences in the spatial pattern between the *aerosol with*  $2xCO_2$  and *aerosol only* simulations. Areas of high orography such as the Western Ghats and the Himalayas experience slightly less rainfall under combined absorbing aerosol and carbon dioxide forcing, than with absorbing aerosol forcing alone (Figure 910, top row). Moreover, the leeward side of the Himalayas receives slightly more rainfall, suggesting a downwind shift of precipitation in response to carbon dioxide induced warming (Siler and Roe, 2014). For absorbing aerosol forcing larger than  $60W/^2$ , in the combination of doubled carbon dioxide concentration, the absolute response of the precipitation is to decrease.

Aerosol with  $2xCO_2$  simulation. Contours showing mean decadal June-July-August precipitation anomaly (top row) & precipitable water anomaly (bottom row), compared to aerosol only run (anomaly = aerosol with  $2xCO_2$  – aerosol only), for a range of aerosol forcing values.

Figure 10. Aerosol with  $2xCO_2$  simulation. Contours showing mean decadal June-July-August 850 hPa precipitation anomaly (top row) & 200 hPa precipitable water anomaly (bottom row) wind speed & direction anomaly, compared to aerosol only run (anomaly = aerosol with  $2xCO_2$  - aerosol only), for a range of absorbing aerosol forcing values. Areas of high orography are masked in greyStippling where the anomaly exceeds double the JJA interannual variability.

**Figure 11.** *Aerosol with 2xCO*<sub>2</sub> simulation. Contours showing mean decadal June-July-August 850 hPa (top row) & 200 hPa (bottom row) wind speed anomaly, compared to *aerosol only* run (anomaly = *aerosol with 2xCO*<sub>2</sub> - *aerosol only*), for a range of absorbing aerosol forcing values. Areas of high orography are masked in grey. Stippling where the anomaly exceeds double the JJA interannual variability.

The effect of the combined forcing is to further weaken the large-scale circulation with respect to the *aerosol only* run. In particular, there is a additional weakening of the low-level southwesterly wind (Figure 1011, top row), reducing the moisture influx from the Arabian Sea to India. At mid-levels (not shown), the anti-cyclonic wind over the Middle East is stronger in the *aerosol with*  $2xCO_2$  simulation than the *aerosol only* simulations, corresponding to an area of high pressure. Figure 10-11 (bottom row), for the high level wind field, shows a further weakening of the Tropical Easterly Jet compared to the *aerosol only* run. In contrastOn the other hand, there is a slight strengthening of the westerly subtropical jet (located between 30-45°N), likely related to the warmer mid-tropospheric temperatures in the *aerosol with*  $2xCO_2$  simulation and associated increase in meridional temperature gradient in the upper troposphere/lower stratosphere.

# 4.1 Relationship between forcing and precipitation




Referring back to Figure 8, the area-averaged convective precipitation in the There are no statistically significant changes in the vertical velocity for the aerosol with  $2xCO_2$  simulation (dashed lines) is slightly higher than in compared to the aerosol only simulation, but otherwise follows a similar pattern, with East China showing the highest sensitivity to the combined forcing. On the other hand, the amount of precipitable water is much greater when the carbon dioxide levels are doubled. The precipitable water increases slightly with increasing aerosol forcing from 0 to  $60W/m^2$  for the aerosol with  $2xCO_2$  run, whilst

the precipitation is monotonically decreasing for all regions except South India. Once again, we highlight the discrepancy between the precipitable water and the precipitation, noting that the elevated levels of precipitable water do not correspond to comparable increases in precipitation as shown by the lack of stippling in Figure S11.

# 440 4.1 Summary & discussion




In general, greenhouse gases act to warm the surface temperature, enabling greater moisture uptake of the atmosphere and leading to enhanced rainfall (e.g. Douville et al. (2000); Ueda et al. (2006); Cherchi et al. (2011); Samset et al. (2018); Cao et al. (2022) Douville et al. (2000); Ueda et al. (2006); Held and Soden (2006); Cherchi et al. (2011); Samset et al. (2018); Cao et al. (2022) ), whilst aerosols are responsible for cooling and drying trends (Monerie et al., 2022; Cao et al., 2022). We find that enhanced carbon dioxide levels act to partially mitigate the effect of imposed absorbing aerosol forcing, although the effects of the absorbing aerosol forcing dominate, in agreement with Cao et al. (2022). Compared to the *aerosol only* simulation, the *aerosol with 2xCO*<sub>2</sub> simulation is warmer at the surface and aloft, has a significantly higher amount of precipitable water and slightly higher precipitation, weaker low-level winds and Tropical Easterly Jet, but a stronger westerly subtropical iet.

We find further evidence that the precipitation decrease in response to aerosol forcing results from a reduction in precipitation efficiency, rather than from a drying of the atmosphere. Looking at Figures 8 and 9, there is a much greater increase in the precipitable water than the precipitation under combined forcing, compared to solely aerosol forcing

**Table 2.** Slopes for Figure 9, taken as a linear fit for the ranges 0–60 and 60–80 W/m<sup>2</sup> absorbing aerosol forcing, for *aerosol with 2xCO*<sub>2</sub> simulation.

|                      |            | e precipitation<br>y per W/m <sup>2</sup> ) | Č                                                         | le precipitation<br>y per W/m <sup>2</sup> ) | 1                       | nble water<br>per W/m <sup>2</sup> ) | Surface temperature (°C per W/m²) |              |  |
|----------------------|------------|---------------------------------------------|-----------------------------------------------------------|----------------------------------------------|-------------------------|--------------------------------------|-----------------------------------|--------------|--|
| Forcing range (W/m²) | 0-60 60-80 |                                             | 0-60                                                      | 60-80                                        | 0-60 60-80              |                                      | 0-60                              | <u>60-80</u> |  |
| North India          | -0.021     | -0.027                                      | 0.001                                                     | 0.009                                        | 0.070                   | -0.047                               | -0.035                            | -0.083       |  |
| South India          | 0.005      | -0.021                                      | 0.003                                                     | 0.009                                        | 0.087                   | -0.014                               | -0.018                            | -0.051       |  |
| East China           | -0.049     | -0.120                                      | 0.006                                                     | 0.007                                        | 0.084                   | -0.111                               | -0.006                            | -0.062       |  |
| Southeast Asia       | -0.076     | -0.009                                      | $\underbrace{0.001}_{00000000000000000000000000000000000$ | $\underbrace{0.008}_{\bullet}$               | $\underbrace{0.042}_{}$ | -0.122                               | $\underbrace{0.010}_{}$           | -0.058       |  |

Referring back to Figure 9, the area-averaged convective precipitation in the *aerosol with*  $2xCO_2$  simulation (dashed lines) is slightly higher than in the *aerosol only* simulation, but otherwise follows a similar pattern, with East China showing the highest sensitivity to the combined forcing and greatest rate change between 0–60 and 60–80W/m² (Table 2). On the other hand, the amount of precipitable water is much greater when the carbon dioxide levels are doubled. The precipitable water increases slightly between 0–60W/m² to 60–80W/m² absorbing aerosol forcing (Table 2), for the *aerosol with*  $2xCO_2$  run. Once again, we highlight the discrepancy between the precipitable water and the precipitation, noting that the elevated levels of precipitable water under combined absorbing aerosol and carbon dioxide forcing do not correspond to comparable increases in convective

precipitation, due to a reduction in the convective precipitation efficiency and the corresponding increase in static stability of the lower troposphere.

Another finding is that the difference between the two simulations, *aerosol with*  $2xCO_2$  and *aerosol only*, depends only weakly on the value of the absorbing aerosol forcing. Considering Figures 9 & 10 10 & 11 (and Figures S7–S11), there is little difference in the three columns, which represent approximate forcings of  $30W/m^2$ ,  $60W/m^2$  and  $90W/m^2$ . This indicates a fairly linear behaviour.

The competition between aerosol and greenhouse gas forcing with respect to the South and East Asian monsoons has been explored in a range of modelling experiments (Samset et al., 2018; Wilcox et al., 2020; Zhou et al., 2020; Ayantika et al., 2021; Swaminathan et al., 2022; Monerie et al., 2022), yet the uncertainty in the forcing itself limits the degree to which the response can be constrained. Our results suggest that in the future, the anticipated reduction in anthropogenic aerosol concentration may have a greater impact on monsoonal precipitation than the increase in greenhouse gases.

# 470 5 Sensitivity to area of applied forcing







The previous experiments (Sections 3 & 4) involved applying a radiative forcing simultaneously across three regions—: India, Southeast Asia and East China (as per Figure 5)—simultaneously. Several aspects of the monsoon response, such as the anomalous summer precipitation over North India, merit further investigation. The method employed here is to apply the radiative forcing to each region separately, to decompose the local and non-local effects of the forcing. This allows better understanding and attribution of the different spatial responses to specific areas of applied forcing. The focus is on an absorbing aerosol forcing value of 60W/m², which corresponds to an intense, yet not physically unreasonable, forcing.

Figure 11 Figure 12 shows the precipitation, surface temperature and wind speed anomalies, with respect to the control run, for the three simulations. The June July August simulations with forcing applied to all regions (first column), India (second column), East China (third column) and Southeast Asia (fourth column). The JJA results are averaged over a 50-year 100-year period. The local effects of applying the absorbing aerosol forcing are similar for each region; namely, cooler surface temperatures and a reduction in precipitation. In terms of non-local effects, there are several contrasting responses. Applying aerosol forcing over India leads to a slight increase in the precipitation in Over East China and Southeast Asia, linked with changes in the large-scale circulation. Similarly, when applying the forcing over Southeast Asia, the changes in precipitation are statistically significant, whilst the local reduction in surface temperature is statistically significant for all regions. Although there is a slight increase reduction in precipitation over East China, due to increased low-level moisture transport from the South China Sea. However, the strongest remote effect is India under locally applied forcing, it does not show as significant due to the high interannual variability of summer rainfall for the Indian region.

The strongest non-local effects are observed when applying forcing over East China. Firstly, there is a significant reduction in precipitation in East Siberia, noted previously in Section 3, and secondly, there is an , which leads to a slight reduction in surface temperature and a slight increase in precipitation over India (Figure 12, top two rows). The precipitation response of India to forcing applied over East China is nearly as strong as when the forcing is applied locally, albeit with opposing trends.

Figure 12. Impact of applying 60W/m<sup>2</sup> absorbing aerosol forcing in turn to regions of India, Southeast Asia and East China. 50-year 100-year June-July-August mean anomaly of labelled variables, compared to the control run. Areas of high orography are masked in grey. Stippling where the anomaly exceeds double the JJA interannual variability.

This results in net change of nearly zero for the precipitation over India at 60W/m<sup>2</sup> forcing, as seen in Figure 6 (top row) and discussed in Section 3. Similar asymmetry in the teleconnection between East China and India in relation to local aerosol forcing had absorbing aerosol forcing has been shown in Herbert et al. (2022). There is a reasonably well-established link between the Indian monsoon and China, via the Silk Road pattern, which is associated with eastward-propagating upper-level Rossby waves (Enomoto et al., 2003; Ding and Wang, 2005; Chakraborty et al., 2014; Boers et al., 2019). However, our case seems to be the reverse, with the effects from forcing over East China propagating westwards towards India. Thus, further study is required to fully understand the underlying mechanisms.



Looking at the surface temperature in Figure 11 (second row), it can be seen that applying forcing independently to East China and to Southeast Asia have opposing effects on the surface temperature over India, with the former acting to cool the

surface and the latter acting to warm the surface. The contrasting impacts on surface temperature cancel out when the regions are forced simultaneously.

In terms of the low-level circulation (Figure 112, third row), applying forcing to India leads to a slight increase in wind speed from North to South the west coast to central India. Applying forcing to Southeast Asia causes a slight reduction in the low-level monsoon flow from the Arabian Sea to India, whilst applying forcing to East China causes a reduction in the low-level flow from the Bay of Bengal to Southeast Asia, and an increase from East China to Northeast China. At high levels (Figure 112, fourth row), forcing Southeast Asia or East China induces an increase in the westerly subtropical jet speed and a decrease in the Tropical Easterly Jet wind speedand an increase in the westerly subtropical jet speed, although this is much more pronounced. The change in the Tropical Easterly Jet is statistically significant when applying the forcing to East China. Generally, the local response to regionally applied absorbing aerosol forcing is consistent with the response to the large-scale forcing (Section 3). India is the most sensitive to remote forcing, with the local precipitation, surface temperature and circulation being affected when forcing is applied to Southeast Asia or East China. We find that for Southeast Asia and East China, applying the absorbing aerosol forcing locally gives the greatest response, whilst for India, the responses to local and remote forcing are comparable. This is in agreement with Sherman et al. (2021) and Chakraborty et al. (2014), who note the importance of both Chinese and Indian emissions on precipitation over India, but in disagreement with Bollasina et al. (2014) and Undorf et al. (2018), who suggest prioritising the influence of local aerosol sources. On the other hand, Guo et al. (2016) find that the biggest contributors to precipitation changes over India are from remote sources. Uncertainty in future emissions

**Table 3.** Values of area-averaged surface temperature (°C) at ~30 & 60W/m<sup>2</sup> absorbing aerosol forcing for different regions (rows) and model simulations (columns). Values taken as 100-year June-July-August averages.

scenarios, in terms of location and intensity, remains a barrier to predicting the response of the South and East Asian monsoons.

|                      | aerosol+all (temperature °C) |           | aerosol+India* (temperature °C) |           | aerosol+E China* (temperature °C) |           | aerosol+SE Asia* (temperature °C) |           | average of * (temperature °C) |           |
|----------------------|------------------------------|-----------|---------------------------------|-----------|-----------------------------------|-----------|-----------------------------------|-----------|-------------------------------|-----------|
| Forcing range (W/m²) | <u>30</u>                    | <u>60</u> | <u>30</u>                       | <u>60</u> | <u>30</u>                         | <u>60</u> | <u>30</u>                         | <u>60</u> | 30                            | <u>60</u> |
| North India          | 29.3                         | 27.3      | 29.6                            | 28.4      | 30.2                              | 29.3      | 31.0                              | 30.8      | 30.3                          | 29.5      |
| South India          | 24.5                         | 23.4      | 24.6                            | 23.8      | 25.0                              | 24.7      | 25.5                              | 25.5      | 25.0                          | 24.7      |
| East China           | 23.0                         | 21.9      | 23.6                            | 23.6      | 23.1                              | 22.4      | 23.5                              | 23.5      | 23.4                          | 23.2      |
| Southeast Asia       | 23.8                         | 22.8      | 24.0                            | 24.1      | 24.0                              | 24.1      | 23.6                              | 23.2      | 23.9                          | 23.8      |

# 5.1 Linearity of the response



**Table 4.** Values of area-averaged precipitation (mm/day) at ~30 & 60W/m<sup>2</sup> absorbing aerosol forcing for different regions (rows) and model simulations (columns). Values taken as 100-year June-July-August averages.

|                                   | aerosol+all (precip. mm/day) |           | aerosol+India* (precip. mm/day) |           | aerosol+E China*<br>(precip. mm/day) |           | aerosol+SE Asia*<br>(precip. mm/day) |           | average of * (precip. mm/day) |     |
|-----------------------------------|------------------------------|-----------|---------------------------------|-----------|--------------------------------------|-----------|--------------------------------------|-----------|-------------------------------|-----|
| Forcing range (W/m <sup>2</sup> ) | <u>30</u>                    | <u>60</u> | <u>30</u>                       | <u>60</u> | <u>30</u>                            | <u>60</u> | <u>30</u>                            | <u>60</u> | 30                            | 60  |
| North India                       | <u>6.1</u> ∼                 | 5.3       | <u>6.0</u>                      | 5.4       | <u>6.7</u>                           | 7.1       | <u>6.2</u>                           | 6.1       | <u>6.3</u>                    | 6.2 |
| South India                       | <u>7.0</u>                   | 7.5       | <u>6.8</u>                      | 6.5       | <del>7.5</del> €                     | 8.6       | <u>6.8</u>                           | 6.8       | 7.0                           | 7.3 |
| East China                        | 7.4                          | 5.2       | <u>8.2</u>                      | 8.3       | 7.2                                  | 5.4       | <u>8.4</u>                           | 8.9       | 7.9                           | 7.5 |
| Southeast Asia                    | 7.9                          | 5.7       | 9.6                             | 9.5       | 9.3                                  | 9.0       | <u>8.4</u>                           | 6.5       | 9.1                           | 8.3 |

Combining the We investigate the response to linearly combining the separately forced regions of India, Southeast Asia and East China, (Section 5) and comparing to the *aerosol only* simulation at a against the response to forcing all regions simultaneously. Absorbing aerosol forcings of intensity 60W/m<sup>2</sup> snapshot, where all the regions are forced simultaneously, there is a fair degree of linearity in the response (see Figure 12). Looking at Figure 12, the left and right columns (medium) and 30W/m<sup>2</sup> (low) are considered, where the latter has been scaled by a factor of two for easier visual comparison.




Looking at Figures 13 & 14, the first two columns (60W/m²) and the first two rows (2x30W/m²), are qualitatively similar. Tables 1 and 2 3 and 4 further quantify the linearity of the response, presenting the regionally averaged June July August JJA surface temperature (Table 13) and regionally averaged June July August JJA precipitation (Table 24). We can compare values from the *aerosol only* simulation with simulation with 60W/m² absorbing aerosol loading across all regions simultaneously, against the simulations where the 60W/m² absorbing aerosol forcing was applied to a single region – India, Southeast Asia or East China – at a time. Generally, the *aerosol only* simulation is slightly cooler with and has The response of simultaneously forced regions is marginally stronger, with slightly cooler surface temperatures and less precipitation than the linear combination of the responses to forcing each region independently. Statistically, there is little difference in the precipitation response between simultaneously forced and linearly combined independently forced regions (shown by lack of stippling in third column of Figure 13). Similarly, there is little difference in the response between 60W/m² and 2x30W/m². In Section 4, we noted that the response of variables between 30, although still comparable. 60 and 90W/m² was comparable. Generally, there is a fair degree of linearity in the response in terms of forcing intensity and combinations of forcing regions.

These results are in contrast to Herbert et al. (2022), who found a non-linear response when North India and East China were forced separately, compared to being forced simultaneously. Their experiments were also conducted using an intermediate complexity climate model, with an approximate treatment of scattering and absorbing aerosols, but considered the effects of

Figure 13. Precipitation, taken as 100-year June-July-August average, for 60W/m² (top row) and surface temperature 30W/m² (bottom row)—absorbing aerosol forcing, where the latter has been scaled by a factor of 2. Left column: 50-year June-July-August mean anomaly (aerosol only—against control) that approximately corresponds to 60W/m² of simultaneously forcing all 3 regions. Right-Middle column: sum anomaly against control of anomalies from regionally the linear combination of separately forced runs regions (India/, East China /SE & Southeast Asia—control), taken as 50-year June-July-August average. Right column: difference between left & middle columns. Stippling where the anomaly exceeds double the JJA interannual variability of the control simulation.

removing them rather than adding them. The differences are likely related to the areas of applied forcing; Herbert et al. (2022) apply aerosol forcing to a limited region along the north boundary of India, whilst we apply aerosol forcing across the entirety of India. The limited region of application by Herbert et al. (2022) may elicit stronger orographic and advective effects, leading to a greater degree of non-linearity in the response. On the other hand, Shindell et al. (2012) noted generally linear responses of summer precipitation to forcing combinations involving aerosols and greenhouse gases, across zonally averaged latitudinal bands. Similarly, on a global scale, Gillett et al. (2004) find no evidence of non-linearity in the combination of responses to greenhouse gases and sulphate aerosols with the HadCM2 model. Guo et al. (2016) find that the reduction in precipitation across Southeast Asia due to higher global sulphur dioxide emissions is comparable to the linear combination of precipitation from simulations that consider local and remote sources of sulphur dioxide independently. However, a similar linearity is not observed in the response to increases in black carbon aerosols. Further work is required to quantify the linearity in response to aerosol forcing across different models, and how model biases can impact the results.

**Figure 14.** Surface temperature, taken as 100-year June-July-August average, for 60W/m<sup>2</sup> (top row) and 30W/m<sup>2</sup> (bottom row) absorbing aerosol forcing, where the latter has been scaled by a factor of 2. Left column: anomaly against control of simultaneously forcing all 3 regions. Middle column: anomaly against control of the linear combination of separately forced regions (India, East China & Southeast Asia). Right column: difference between left & middle columns. Stippling where the anomaly exceeds double the JJA interannual variability of the control simulation.

#### 6 Conclusions



The strength of the South and East Asian monsoons is largely determined by processes affecting the sea surface temperature, greenhouse gases and aerosols. Here, we have conducted a parametric study with an intermediate complexity climate model, to assess the roles of absorbing aerosol and greenhouse gas forcing on the South and East Asian monsoons. In addition, we have identified the level of regional forcing at which the monsoon system breaks down, in terms of a significant reduction in precipitation.

Absorbing aerosol forcing, which we apply through a combination of mid-tropospheric heating and surface cooling in our model, causes decreasing surface temperatures, mid-level warming, weakening circulation and a reduction in (convective) precipitation. Surprisingly, as As the forcing increases, the precipitation declines much faster than the precipitable water, indicating that it is due to a lack of precipitation efficiency related to changes in the stratification of the atmosphere, rather than due to a lack of moisture. Advection of dry air from East China leads to a reduction in precipitation in eastern Siberia, which is outside of the area being forced. On removal of the absorbing aerosol forcing, we find that the monsoon system recovers fully, indicating that there is no hysteresis in our model simulations. Doubling carbon dioxide concentration partially

mitigates the effects of the <u>absorbing</u> aerosol forcing, through warmer surface temperatures enabling greater moisture takeup, but further weakens the large-scale circulation. <del>On removal of the aerosol forcing, we find that the monsoon system recovers fully, indicating that there is no hysteresis in our model simulations. We find that, when considering realistic ranges of applied forcings, the precipitation response is more sensitive to absorbing aerosol than greenhouse gas forcing, highlighting the importance of air quality policies and the impact they can have on the future state of the South and East Asian monsoons.</del>







The strongest regional responses, particularly in regards to the circulation, are attributed to absorbing aerosol loading over East China. Although the precipitation decline for each region directly corresponds to applying forcing to that region, there is a remote connection between East China and India. Forcing applied to East China leads to an a slight increase in precipitation over India, which is in contrast to the response when forcing is applied to India. When both regions are forced simultaneously, there is a reduction in precipitation over India, but the reduction is much less than for Southeast Asia or East China. Comparing simulations where the regions have been forced separately to the simulation where the regions have been forced simultaneously, the results are qualitatively similar, indicating a fair degree of linearity in the response.

We have characterised regional behavioural regimes in terms of area-averaged precipitation, precipitable water and surface temperature. India is separated into North and South regions, due to the significant variance in their responses. South India is the least affected region, likely due to its peninsula nature. North India shows an approximately linear decrease in precipitation in relation to the acrosol forcing. The precipitation response of North India and Southeast Asia to increasing absorbing acrosol forcing is an approximately linear decline, with Southeast Asia showing a stronger negative sensitivity than North India. For East China, there is an sharp transition at around 60W/m² to a regime where the precipitation is close to zero, indicting indicating tipping behaviour. The precipitation response of Southeast Asia is somewhere between the other regions, with precipitation declining at a faster rate than North India but without the abrupt transition of East China. In terms of the precipitable water, it remains relatively constant for all regions until 60W/m²; thereafter the precipitable water linearly declines with further increases in forcing. The surface temperature behaves similarly to the precipitable water, but becomes much more sensitive to the forcing beyond 60W/m² and declines non-linearly.

We note the importance of aerosol loading over East China and the competition between aerosol loading over India, in determining the response of the Indian region to future climate scenarios. At approximately 60W/m² of absorbing aerosol forcing, there is a clear transition from a high to a low precipitation regime for the East China region, whilst Southeast Asia and South India show a more gradual linear decline in precipitation with increasing forcing. There is a compensating effect from East China absorbing aerosol forcing on precipitation over India, resulting in a lesser decline of precipitation over India compared with Southeast Asia or East China. For area-averaged precipitable water and surface temperature, there is transition at around 60W/m² for all regions from near-constant to decreasing levels. To maintain a safe operating space for the South and East Asian monsoons, it is suggested to keep our results suggest keeping the absorbing aerosol forcing below 60W/m², through air quality policies and collaboration between Asian countries.

Our results are limited by the low resolution and lack of explicit aerosol interactions and chemistry, but future work will aim to address these issues by repeating similar experiments using a more complex global climate model. We would consider the effects of scattering aerosols as well as absorbing aerosols, both in combination and in isolation. Additionally, we would like

to investigate the impact of applying forcing over a shorter seasonal period rather than perennially, particularly with regards to pre-monsoon conditions and frequency of extreme weather events.

*Code availability.* Model version used in this paper is available on GitHub (not including the aerosol forcing modifications): https://github.com/jhardenberg/PLASIM.

Data availability. Model data used to make the figures in this paper is publicly available on Figshare: https://figshare.com/projects/Supplementary\_material\_for\_Recchia\_et\_al\_2022/142400.

Author contributions. LGR & VL wrote the paper. VL designed the simulations and LGR and performed the analysis.

Competing interests. The authors declare that they have no conflict of interest.


Acknowledgements. This project is TiPES contribution #186. This project has received funding from the European Union's Horizon 2020 research and innovation programme under grant agreement No. 820970. We thank Frank Lunkeit, Shabeh ul Hasson and Milind Mujumdar for their contributions and insight. We thank T. Stocker for suggesting this line of research, and also the three reviewers for their constructive comments which have substantially improved this manuscript.

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

## **Supplementary figures**

**Figure S1.** *Aerosol only* simulation. Contours showing mean decadal June-July-August temperature and mean decadal June-July-August temperature and mean decadal June-July-August temperature anomaly compared to the control run (anomaly = aerosol only - control), for a range of aerosol forcing values. The top two rows are at the surface and the bottom two rows at 700 hPa. Areas of high orography are masked in grey. Stippling where the anomaly exceeds double the JJA interannual variability.

**Figure S2.** *aerosol only Aerosol only* simulation. Contours showing mean decadal June-July-August <u>evaporation</u> 500 hPa geopotential height and mean decadal June-July-August <u>evaporation</u> 500 hPa geopotential height anomaly compared to the control run (anomaly = *aerosol only* - control), for a range of aerosol forcing values. Areas of high orography are masked in grey. Stippling where the anomaly exceeds double the JJA interannual variability.

**Figure S3.** *Aerosol only* simulation. Contours showing mean decadal June-July-August 500 hPa geopotential height (top left) evaporation and mean decadal June-July-August 500 hPa geopotential height evaporation anomaly compared to the control run (anomaly = aerosol only - control), for a range of aerosol forcing values. Areas of high orography are masked in greyStippling where the anomaly exceeds double the JJA interannual variability.

Figure S4. Aerosol with 2xCO<sub>2</sub> Aerosol only simulation. Contours showing mean decadal June-July-August surface-Vertical section along 20°N (top rowtwo rows) & 700 hPa and 109°E (bottom rowtwo rows) with contours showing mean decadal June-July-August temperature & temperature anomaly, compared to aerosol only control run (anomaly = aerosol with 2xCO<sub>2</sub> – aerosol only - control), for a range of aerosol forcing values. Areas of high orography are masked in grey. Stippling where the anomaly exceeds double the JJA interannual variability.

**Figure S5.** *Aerosol only* simulation. Vertical section along 20°N (top two rows) and 109°E (bottom two rows) with contours showing mean decadal June-July-August specific humidity & specific humidity anomaly, compared to control run (anomaly = *aerosol only* - control), for a range of aerosol forcing values. Areas of high orography are masked in grey. Stippling where the anomaly exceeds double the JJA interannual variability.

Figure S6. Aerosol only simulation. Vertical section along  $20^{\circ}$ N (top two rows) and  $109^{\circ}$ E (bottom two rows) with contours showing mean decadal June-July-August vertical velocity ( $\omega$ ) & vertical velocity anomaly, compared to control run (anomaly = aerosol only - control), for a range of aerosol forcing values. Dotted lines show the convective precipitation/convective precipitation anomaly along the section. Areas of high orography are masked in grey. Stippling where the anomaly exceeds double the JJA interannual variability.

**Figure S7.** Aerosol with  $2xCO_2$  simulation. Contours showing mean decadal June-July-August surface (top row) & 700 hPa (bottom row) temperature anomaly, compared to aerosol only run (anomaly = aerosol with  $2xCO_2$  - aerosol only), for a range of aerosol forcing values. Areas of high orography are masked in grey. Stippling where the anomaly exceeds double the JJA interannual variability.

**Figure S8.** Aerosol with  $2xCO_2$ . Contours showing mean decadal June-July-August 925 hPa (top row) & 700 hPa (bottom row) specific humidity anomaly, compared to *aerosol only* run (anomaly = aerosol with  $2xCO_2$  - aerosol only), for a range of aerosol forcing values. Areas of high orography are masked in grey. Stippling where the anomaly exceeds double the JJA interannual variability.

**Figure S9.** Aerosol with  $2xCO_2$  simulation. Vertical section along  $20^{\circ}$ N (top two rows) and  $109^{\circ}$ E (bottom two rows) with contours showing mean decadal June-July-August temperature anomaly, compared to *aerosol only* run (anomaly = *aerosol with*  $2xCO_2$  - *aerosol only*), for a range of aerosol forcing values. Areas of high orography are masked in grey. Stippling where the anomaly exceeds double the JJA interannual variability.

**Figure S10.** Aerosol with  $2xCO_2$  simulation. Vertical section along  $20^\circ$ N (top two rows) and  $109^\circ$ E (bottom two rows) with contours showing mean decadal June-July-August specific humidity anomaly, compared to aerosol only run (anomaly = aerosol with  $2xCO_2$  - aerosol only), for a range of aerosol forcing values. Areas of high orography are masked in grey. Stippling where the anomaly exceeds double the JJA interannual variability.

**Figure S11.** Aerosol with  $2xCO_2$  simulation. Vertical section along  $20^\circ$ N (top row) and  $109^\circ$ E (bottom row) with contours showing mean decadal June-July-August vertical velocity ( $\omega$ ) & vertical velocity anomaly, compared to aerosol only run (anomaly = aerosol with  $2xCO_2$  - aerosol only), for a range of aerosol forcing values. Dotted lines show the convective precipitation/convective precipitation anomaly along the section. Areas of high orography are masked in grey. Stippling where the anomaly exceeds double the JJA interannual variability.