# Peer review of "Modelling the effect of aerosol and greenhouse gas forcing on the South & East Asian monsoons with an intermediate complexity climate model"

_EGUsphere, 2023_

## Referee Comment (RC1)

Using an intermediate complexity climate model, Recchia and Lucarini investigate how the Asian monsoon responds to absorbing aerosols. They perform perturbations of increasing magnitude, and they also look at the individual sensitivities to perturbations in three Asian subregions as well as the linearity in these latter simulations. Additional simulations are performed to study the effect of simultaneous aerosol and CO2 increases. They find reduced summertime precipitation in the regions of applied forcing, approximately linear with the magnitude of the forcing, and a more variable response in surrounding regions. CO2 acts to partially offset the noted precipitation reduction.

With the large ongoing and potentially future emission changes in the Asian region, it is more important than ever to understand how the Monsoon responds to such a forcing. While e.g. Earth System Models include more of the processes potentially at play, the reduced-complexity model used here have the strength of increased transparency and can be easier to interpret. I therefore believe that, given some edits noted below, this paper can be a good contribution to the field and may be fit for publication in Earth System Dynamics.

**General comments:**

The paper is well written, starting with a good presentation of methods data and cases. Results are clearly presented with a fluent language and well-designed figures. I have two general comments, that will recur in the specific comments below:
1) authors need to be clear that their "aerosol forcing" emulates absorbing aerosols only, not scattering aerosols. This should be evident if not in the title, then at least in the abstract and throughout the text.
2) the authors present many good figures, but there is a general lack of quantification and statistics.

**Specific comments:**

P2: both in terms of the response of the monsoons and in the future climate forcing.
Could you please consider rewording this sentence as it's a bit confusing. How the monsoon evolves in a changing climate is uncertain because we don't know enough about the how the monsoon reacts to a forcing. We also don't know exactly how the future forcing of the climate will be. Was that the point?

P2: The multi-model mean usually performs better than any single model [14–17].
Does this have some physical explanation or is it just pure luck?

P3: Generally, aerosols have a stabilising effect on the atmosphere, through surface cooling and mid-tropospheric warming, increasing the stratification of the atmosphere and causing a drying trend [35–37]
This is indeed an important effect, but I disagree that is a *general* effect. This is an example of places in the text where the fact that authors are focusing on absorbing aerosols needs to be made clear.

P3: but moving forward, greenhouse gas forcing is expected to dominate, which is associated with a likely increase in monsoonal rainfall in the northern hemisphere
I agree that aerosols have played a stronger role historically than they will in the future. However, in the near-term we may see quite strong aerosol trends (reductions, presumably, but still) in certain regions. In the longer terms, greenhouse gases will indeed dominate, but I think the authors should consider adding the point that aerosols are still important in the near-term (also elevating the importance of your analyses).

P4: The main goal is to understand the impact of aerosol forcing on the
Please add "absorbing" before "aerosols".

P6: Here, we use a combination of our own simulations with the PLASIM model and results from existing literature that use a hierarchy of models to quantify the responses of the South and East Asian monsoons to a range of future climate scenarios
It is not entirely clear what is meant here: are results from other studies used in this study? If you just refer to

other studies in the text it would be good to reword this sentence so it doesn't seem like they are a direct part of this study.

P7: Figures1–4 show the performance of a 50-year control simulation with the PLASIM model
I usually think of a model's performance as how the model does compared to the real world (i.e., compared to observations). I realize that you show and explain how the model does display real-world features, but section 2.1 in general lacks the observation element. It is obvious that the authors know their field and the description of the PLASIM features in the text is excellent, but the reader needs to be shown – not just told, without any references – that it agrees with observations. Adding references to the literature when describing different dynamical features, as well as a figure or two comparing PLASIM directly with observations is needed here.

P10: The added aerosol forcing causes the surface to cool
It would be very interesting to see a regional average of the vertical temperature profile, to see how it changes with this "aerosol forcing".

Figure 7, arrow sizes: In the figures with wind arrows, it is extremely difficult to see the direction of the arrows. Would it be possible to play with the plotting here, trying e.g. to make the arrowheads larger, to have fewer but larger arrows, or something like that?

Figure 7, arrow directions and wind speed anomalies: I'm having difficult with these winds, and I would like to underline that if there is a well established consensus that this is how wind changes are displayed, then the following comment may be disregarded: Authors have chosen to show anomalies in both wind speed and in wind direction. I'm not sure this is the best way to convey the results, and to show how the climate of the region looks under a strong aerosol-like perturbation. For instance, looking at the 850hPa panel for ~150W/m2, there is a strong blue band stretching across south India and towards the southwest. This dark blue color should be read as: the SW monsoon wind has weakened dramatically and is close to zero. This, to me, is very counter intuitive. Also, when arrows point in the opposite direction, this does not necessarily signify that the average wind direction has turned? It would be easier to interpret the changes if maps showed absolute wind speeds and directions, so they could be compared directly to the first green maps. Lighter/darker colors than that "baseline" map would mean stronger/weaker winds, and arrows would point in actual wind directions.

P13: There is a strengthening of the low-level southwesterly wind in East China, causing dry air to be advected towards East Siberia
I'm probably misreading this but: a stronger wind from southwest sends dry wind *towards* the west?

P14: We note again that between 0 and60W/m2forcing, although the convective precipitation is gradually reducing,the precipitable water does not correspondingly decline.
Why was that?

P15: A more quantitative angle
This paragraph is one place where I believe the authors could have been more quantitative in their analyses. Could you try to put a quantitative number on the "sensitivity to aerosol forcing", for instance? The numbers would be contrasting nicely between convective precip. and precipitable water.

P15: Regarding the analysis of forcing vs regional climate impact: it would be good to see a few concluding sentences reminding the reader why these results are relevant – what are their link to the real world (small/large emission changes)?

P16: Although the precipitable water is considerably greater when carbondioxide levels are doubled
Provide the reader with a quick explanation of why that is.

P17: noting that the elevated levels of precipitable water do not correspond to comparable increases in precipitation
Please provide a physical explanation to this.

P17: Considering Figures 9&10, there is little difference in the three columns, which represent approximate forcings of 30W/m2, 60W/m2 and 90W/m2. This indicates a fairly linear behaviour.
These linearities must be quantified. Spatial correlations between the maps are one suggestions, but I'm sure there are other.

P17: Our results suggest that in the future, the anticipated reduction in aerosol concentration may have a greater impact on monsoonal precipitation than the increase in greenhouse gases.
This is a nice and clear result, written in a way that conveys the relevance of this study. This message should be underlined – in the abstract and/or in the final conclusion.

P18: The precipitation response of India to forcing applied over East China is nearly as strong as when the forcing is applied locally, albeit with opposing trends.
Again, please provide a physical explanation, or at least a suggestion to one.

P20: On removal of the aerosol forcing, we find that the monsoon system recovers fully, indicating that there is no hysteresis in our model simulations.
Where is this shown? The reader needs to see this finding.

Conclusion: Given the tool used,  I believe the method of emulating absorbing aerosols is as good as any. However, it would be good to see a short discussion of caveats connected to the very idealized nature of this type of perturbation.

**Technical corrections:**

Line numbers on the document will greatly ease the reviewer job in the next round!

First sentence of introduction: "region economy" → "region's economy"

P14: a little bug: ¡60W/m2

P14: form → from: form 0 to 150W/m2

P14: sdepends → depends with 2xCO2andaerosol only, sdepends

---

## Author Comment (AC1)

**Response to reviewer 1**

Format of responses: (1) comments from referees/public, (2) author's response, and (3) author's changes in the manuscript.

**General comments:**

(1) The paper is well written, starting with a good presentation of methods data and cases. Results are clearly presented with a fluent language and well-designed figures. I have two general comments, that will recur in the specific comments below:
a) authors need to be clear that their "aerosol forcing" emulates absorbing aerosols only, not scattering aerosols. This should be evident if not in the title, then at least in the abstract and throughout the text.
b) the authors present many good figures, but there is a general lack of quantification and statistics.
(2) We thank the reviewer for their comments and for taking the time to review the manuscript. We have now clarified that we are primarily concerned with the emulation of absorbing aerosols and their effects on the Asian monsoons. Furthermore, we have improved the statistical analysis of our results (details below).
(3) Various instances in the text have been updated to specifically refer to absorbing aerosols. Figures have been updated with stippling to show statistically significant changes. These are determined as where the anomaly (e.g. 60W/m$^2$ forcing – control) is greater than 2 times the interannual variability of the June-July-August mean, with respect to the reference simulation (e.g. 60W/m$^2$ forcing). In conjunction, the text has edited to include discussion of the statistical relevance of the anomaly signals. Example of figures with added stippling can be seen in response to some of the comments below.

**Specific comments:**

(1) P2: both in terms of the response of the monsoons and in the future climate forcing.
Could you please consider rewording this sentence as it's a bit confusing. How the monsoon evolves in a changing climate is uncertain because we don't know enough about the how the monsoon reacts to a forcing. We also don't know exactly how the future forcing of the climate will be. Was that the point?
(2) Yes – firstly, that response of the monsoon to a forcing is uncertain, and secondly that the nature of the forcing itself is uncertain. Reworded sentence to clarify.
(3) "There remains considerable uncertainty in how the monsoons will evolve in a changing climate, both in terms of the response of the monsoons to an applied forcing and in the nature of the future climate forcing itself."

(1) P2: The multi-model mean usually performs better than any single model [14–17]. Does this have some physical explanation or is it just pure luck?
(2) The use of multi-model ensembles is widespread and it is often assumed or found that the multi-model average performs better than any model. The argument is that by averaging over different models, structural uncertainty is partly taken care of (Tebaldi and Knutti, 2007: https://royalsocietypublishing.org/doi/10.1098/rsta.2007.2076 ). However, as one of the authors has discussed in Ghil and Lucarini, 2020 (https://doi.org/10.1103/RevModPhys.92.035002), this is far from being a scientifically error-proof approach. What one can say, instead, is that the use of multiple models allows for a wider spread than a single-model ensemble, hence more thoroughly sampling model error. It is really not the goal of this paper to dwell on this topic, despite the evident importance.
(3) No changes implemented.

(1) P3: Generally, aerosols have a stabilising effect on the atmosphere, through surface cooling and mid- tropospheric warming, increasing the stratification of the atmosphere and causing a drying trend [35–37]

This is indeed an important effect, but I disagree that is a *general* effect. This is an example of places in the text where the fact that authors are focusing on absorbing aerosols needs to be made clear.

(2) We have removed "mid-tropospheric warming" so that the sentence can refer to aerosols in general, with the following paragraph going into more detail regarding absorbing and scattering aerosols and their specific effects.

(3) "Generally, aerosols have a stabilising effect on the atmosphere, through surface cooling, increasing the stratification of the atmosphere and causing a drying trend which results in a weaker monsoon (Li et al., 2016; Wilcox et al., 2020; Ayantika et al., 2021; Cao et al., 2022)."

(1) P3: but moving forward, greenhouse gas forcing is expected to dominate, which is associated with a likely increase in monsoonal rainfall in the northern hemisphere

I agree that aerosols have played a stronger role historically than they will in the future. However, in the near- term we may see quite strong aerosol trends (reductions, presumably, but still) in certain regions. In the longer terms, greenhouse gases will indeed dominate, but I think the authors should consider adding the point that aerosols are still important in the near-term (also elevating the importance of your analyses).

(2) Added an extra sentence concerning importance of aerosols on the Asian monsoons in the near-future. Of course, considerable uncertainties remain with regards to future technological development and implementation (or not) of air quality policies.

(3) "Historically, aerosol forcing has dominated, linked with a declining rainfall trend over the latter half of 20th century (Bollasina et al., 2011; Polson et al., 2014; Li et al., 2015; Undorf et al., 2018; Dong et al., 2019), but moving forward, greenhouse gas forcing is expected to dominate, which is associated with a likely increase in monsoonal rainfall in the northern hemisphere (Monerie et al., 2022). In the near-term, fluctuations of aerosol concentrations in the Asian region will continue to impact the monsoons, with increases in anthropogenic emissions acting to weaken the large-scale circulation and hydrological cycle, thus weakening the monsoons, whilst decreases in emissions, perhaps from air quality policies, will likely intensify the monsoons."

(1) P4: The main goal is to understand the impact of aerosol forcing on the

Please add "absorbing" before "aerosols".

(2) We agree with the reviewer that this needs to be made clearer.

(3) Changed various instances of "aerosols" to "absorbing aerosols".

(1) P6: Here, we use a combination of our own simulations with the PLASIM model and results from existing literature that use a hierarchy of models to quantify the responses of the South and East Asian monsoons to a range of future climate scenarios.

It is not entirely clear what is meant here: are results from other studies used in this study? If you just refer to other studies in the text it would be good to reword this sentence so it doesn't seem like they are a direct part of this study.

(2) Re-worded to make clear that we are using results from a single model, with comparisons to other studies in the text.

(3) "We use the intermediate complexity model, PLASIM, to quantify the responses of the South and East Asian monsoons to a range of future climate scenarios, thereby contributing to the existing literature and ensuring that a hierarchy of models are represented."

(1) P7: Figures1–4 show the performance of a 50-year control simulation with the PLASIM model
I usually think of a model's performance as how the model does compared to the real world (i.e., compared to observations). I realize that you show and explain how the model does display real-world features, but section 2.1 in general lacks the observation element. It is obvious that the authors know their field and the description of the PLASIM features in the text is excellent, but the reader needs to be shown – not just told, without any references – that it agrees with observations. Adding references to the literature when describing different dynamical features, as well as a figure or two comparing PLASIM directly with observations is needed here.

(2) We agree with the reviewer that this should have been discussed in greater detail. Comparison to observations was done at an early stage but not all figures/analysis were included in the manuscript due to concerns over length. Given the idealistic nature of the experiments, extensive quantification of performance against, for example, satellite data, isn't applicable. Our requirements are that PLASIM reproduces the large-scale features of the monsoons, with a clear seasonal cycle, which is hopefully now evident.

(3) We have added subplots to Figures 1-4 to show the equivalent variables from ERA5 reanalysis data, to allow for visual comparison with the PLASIM model. Updated *"Section 1.2 Model Validation"* to reference and discuss the updated figures. Example of revised Figure 1 below.

[Figure]

(1) P10: The added aerosol forcing causes the surface to cool
It would be very interesting to see a regional average of the vertical temperature profile, to see how it changes with this "aerosol forcing".

(2) Figure 8 shows the area-averaged surface temperature against aerosol forcing, and Figures S1 & S4 show the surface & 700 hPa contours for aerosol-only and aerosol-with-2x$CO_2$ forcing. Additionally, we will add latitudinal and longitudinal cross-sectional figures to show the vertical temperature profile changes with aerosol forcing.

[Figure]

(3) Example of a section at 109°E. Aerosol forcing is added approximately between 550 & 750 hPa, corresponding to the warm (red) temperature anomaly at 40-25°N. Surface cooling (blue) is also evident in the same region.

JJA: temperature (°C) at 109°E

JJA: temperature anomaly (°C) at 109°E

(1) Figure 7, arrow sizes: In the figures with wind arrows, it is extremely difficult to see the direction of the arrows. Would it be possible to play with the plotting here, trying e.g. to make the arrowheads larger, to have fewer but larger arrows, or something like that?
(2) It is an ongoing challenge to find the right balance between size, colour and density of the vectors (arrows). We have tried to improve on this, as summarised and shown below.
(3) Reduced number of arrows & made arrowheads larger (example figure 7a below).

[Figure]

JJA: wind speed (m/s) at 850 hPa level

JJA: wind speed anomaly (m/s) at 850 hPa level

(1) Figure 7, arrow directions and wind speed anomalies: I'm having difficult with these winds, and I would like to underline that if there is a well established consensus that this is how wind changes are displayed, then the following comment may be disregarded: Authors have chosen to show anomalies in both wind speed and in wind direction. I'm not sure this is the best way to convey the results, and to show how the climate of the region looks under a strong aerosol-like perturbation. For instance, looking at the 850hPa panel for ~150W/m2, there is a strong blue band stretching across south India and towards the southwest. This dark blue color should be read as: the SW monsoon wind has weakened dramatically and is close to zero. This, to me, is very counter intuitive. Also, when arrows point in the opposite direction, this does not necessarily signify that the average wind direction has turned? It would be easier to interpret the changes if maps showed absolute wind speeds and directions, so they could be compared directly to the first green maps. Lighter/darker colors than that "baseline" map would mean stronger/weaker winds, and arrows would point in actual wind directions.

(2) As above, it is difficult to strike the right balance when displaying vectors, particularly as visual aspects are subjective. Following the reviewers comment, the arrows on the anomaly plots have been removed so as to focus on the changes in wind speed. Representation of the arrows on the forcing (i.e. not anomaly) plots have been improved as per previous comment.

(3) Removed arrows on wind anomaly plots for clarity, as per example figure above.

(1) P13: There is a strengthening of the low-level southwesterly wind in East China, causing dry air to be advected towards East Siberia

I'm probably misreading this but: a stronger wind from southwest sends dry wind towards the west?

(2) A southwesterly wind refers to a wind coming from the southwest and travelling towards the northeast. Edited the relevant sentence in the manuscript to clarify.

(3) "There is a strengthening of the low-level wind in East China (20-40°N), causing more dry air to be advected from southwest to northeast, towards Eastern Siberia, and corresponding to a reduction of precipitation in the region."

(1) P14: We note again that between 0 and 60 W/m2 forcing, although the convective precipitation is gradually reducing, the precipitable water does not correspondingly decline.
Why was that?
(2) There is an explanatory sentence in the previous section: "Therefore, the decline in rainfall is primarily attributed to a reduction in precipitation efficiency, which is due to the increase in the static stability of the lower levels of the atmosphere, as opposed to a scarcity of moisture availability." However, the explanation bears repeating in later sections where necessary.
(3) Added an extra sentence: "This is attributed to a reduction in the precipitation efficiency."

(1) P15: A more quantitative angle (Figure 8)
This paragraph is one place where I believe the authors could have been more quantitative in their analyses. Could you try to put a quantitative number on the "sensitivity to aerosol forcing", for instance? The numbers would be contrasting nicely between convective precip. and precipitable water.
(2) We agree with the reviewer. We have added explicit reference to the sensitivity of convective/large-scale precipitation, precipitable water and surface temperature, to the aerosol radiative forcing in the linear range 0-60 W/m$^2$, and (approximately linear) range 60-80 W/m$^2$. The text in this section already discusses the changes in gradient before and after 60 W/m$^2$, but now we quantify this using the table below, which gives the slopes of the regions/variables following Figure 8.
(3) We have added a table for each simulation to provide quantitative information in support of the text – example of table for *aerosol only* simulation below.

*Table 1: Slopes for Figure 8, taken as a linear fit for the ranges 0-60 and 60-80 W/m$^2$ absorbing aerosol forcing, for* aerosol only *simulation.*

| | Convective precipitation (mm/day per W/m$^2$) | | Large-scale precipitation (mm/day per W/m$^2$) | | Precipitable water (kg/m$^2$ per W/m$^2$) | | Surface temperature (°C per W/m$^2$) | |
|---|---|---|---|---|---|---|---|---|
| Forcing range (W/m$^2$) | 0-60 | 60-80 | 0-60 | 60-80 | 0-60 | 60-80 | 0-60 | 60-80 |
| North India | -0.019 | -0.054 | 0.005 | 0.008 | 0.008 | -0.078 | -0.060 | -0.069 |
| South India | -0.001 | -0.008 | 0.009 | 0.007 | 0.009 | -0.027 | -0.032 | -0.042 |
| East China | -0.058 | -0.115 | 0.008 | 0.008 | 0.005 | -0.154 | -0.024 | -0.103 |
| Southeast Asia | -0.068 | -0.063 | 0.006 | 0.023 | -0.030 | -0.086 | -0.017 | -0.117 |

P15: Regarding the analysis of forcing vs regional climate impact (Section 3.2): it would be good to see a few concluding sentences reminding the reader why these results are relevant – what are their link to the real world (small/large emission changes)?
(2) We agree with the reviewer that it is important to discuss the relevance of the results. But we believe that sufficient discussion is already presented in the Introduction and in the Conclusions. A sentence has been added in Section 3.2 to highlight the relevance of the results.
(3) "East China shows the highest sensitivity to absorbing aerosol forcing of approximately 60W/m$^2$, where the convective precipitation drops to almost zero; essentially a breakdown of the monsoon

system. In terms of future climate scenarios, the aerosol forcing over East China is key for determining the precipitation response, both locally and remotely."

P16: Although the precipitable water is considerably greater when carbon dioxide levels are doubled
Provide the reader with a quick explanation of why that is.
(2) The explanation is given the sentence below. Additionally, we add a reference to Held and Soden, 2006 (https://doi.org/10.1175/JCLI3990.1).
(3) "Given the warmer surface temperatures in the *aerosol with 2xCO$_2$* simulation, we would expect to see greater atmospheric moisture content and higher rates of evaporation and precipitation, compared to the *aerosol only* simulation. Although the precipitable water is considerably greater when carbon dioxide levels are doubled…"

P17: noting that the elevated levels of precipitable water do not correspond to comparable increases in precipitation
Please provide a physical explanation to this.
(2) As per previous comment for P14: the decline is precipitation is due to a reduction in the precipitation efficiency, which results from increased static stability of the lower troposphere. Edited the relevant sentence to add explicit reference to convective precipitation and static stability, as below.
(3) "Once again, we highlight the discrepancy between the precipitable water and the precipitation, noting that the elevated levels of precipitable water do not correspond to comparable increases in convective precipitation, due to a reduction in the convective precipitation efficiency and the corresponding increase in static stability of the lower troposphere."

P17: Considering Figures 9 & 10, there is little difference in the three columns, which represent approximate forcings of 30W/m2, 60W/m2 and 90W/m2. This indicates a fairly linear behaviour
These linearities must be quantified. Spatial correlations between the maps are one suggestions, but I'm sure there are other.
(2) The linearity between the intensity of the forcing and the response to key variables (precipitation, precipitable water & surface temperature) is considered in Figure 8 and Section 3.1. Although linearity in the response is evident from Figure 8 & earlier figures, the focus in Section 3.1 is on identifying regime transitions under extreme forcing. From Figure 8, the response to forcing in the range 0-60W/m$^2$ is reasonably linear; thereafter, we observe changes in the response relative to the applied forcing.
The linearity of the response to area of applied forcing is considered in Section 5.1, with Figure 12 showing the spatial fields. This section has been edited to include results from aerosol forcing of intensity 30W/m$^2$, as well 60W/m$^2$.
(3) Edited text and updated figures (see below) in Section 5.1 to show linearity of the response to forcing at 30W/m$^2$ and 60W/m$^2$. Stippling has been added where the anomaly is greater than 2 times the interannual variability with respect to the control run. By the right column, there are no significant differences in the precipitation response to forcing regions simultaneously or forcing regions independently and combining, indicating linearity in the response. Similarly, the bottom row indicates linear behaviour in the response to the intensity of forcing, 30 vs 60W/m$^2$.

[Figure]

P17: Our results suggest that in the future, the anticipated reduction in aerosol concentration may have a greater impact on monsoonal precipitation than the increase in greenhouse gases.
This is a nice and clear result, written in a way that conveys the relevance of this study. This message should be underlined – in the abstract and/or in the final conclusion.
(2) We agree that this is an interesting result. We don't want to overstate it because it has been obtained in a rather simplified model setting. Having established some interesting results with an intermediate complexity climate model, we believe is it worth pursuing the research using more comprehensive modelling tools.
(3) Added sentences to the Abstract ("These results suggest that in the future, the anticipated reduction in aerosol concentration may have a greater impact on monsoonal precipitation than the increase in greenhouse gases.") and Conclusions ("We find that the precipitation response is more sensitive to absorbing aerosol than greenhouse gas forcing, highlighting the importance of air quality policies and the impact they can have on the future state of the South and East Asian monsoons.") regarding the importance of aerosol vs greenhouse gas forcing. Also added another sentence to the

Conclusions to highlight the limitations of our study, and our future work aims: "Our results are limited by the low resolution and lack of explicit aerosol interactions and chemistry, but future work will aim to address these issues by repeating similar experiments using a more complex global climate model."

P18: The precipitation response of India to forcing applied over East China is nearly as strong as when the forcing is applied locally, albeit with opposing trends.
Again, please provide a physical explanation, or at least a suggestion to one.
(2) The fact that the presence of an aerosol forcing over East China leads to an increase of precipitation over Northern India is an interesting phenomenon. Partly comparable results had been found in Herbert et al. 2022. Understanding this processes is highly nontrivial and we are collaborating with the Herbert et al. team exactly to discover the mechanisms in action.
(3) "The precipitation response of India to forcing applied over East China is nearly as strong as when the forcing is applied locally, albeit with opposing trends. Similar asymmetry in the teleconnection between East China and India in relation to local absorbing aerosol forcing has been shown in Herbert et al. (2022); however, further is required to fully understand the underlying mechanisms."

P20: On removal of the aerosol forcing, we find that the monsoon system recovers fully, indicating that there is no hysteresis in our model simulations.
Where is this shown? The reader needs to see this finding.
(2) Our model is, by construction, rather weak in representing long climatic time scales, given the lack of an active ocean featuring slow dynamics. We add a figure below for the interest of the reviewer, showing the decadal mean precipitation and surface temperature over the length of the simulation (900 years). The start and end years correspond to 0 W/m$^2$ absorbing aerosol forcing, whilst the middle year (450) corresponds to 150 W/m$^2$ absorbing aerosol forcing.
(3) No changes implemented.

[Figure]

(1) Conclusion: Given the tool used, I believe the method of emulating absorbing aerosols is as good as any. However, it would be good to see a short discussion of caveats connected to the very idealized nature of this type of perturbation.

(2) We agree with the reviewer that this is an important point, which we have discussed in detail in Section 2.2. We have also added a sentence to the Conclusions (see below).

(3) As per reviewer comment for P17, added a sentence to the Conclusions: "Our results are limited by the low resolution and lack of explicit aerosol interactions and chemistry, but future work will aim to address these issues by repeating similar experiments using a more complex global climate model."

**Technical corrections:**

(1) Line numbers on the document will greatly ease the reviewer job in the next round!

(2) Sorry - there are line numbers on the submitted article (.pdf) to Earth System Dynamics, but not on the preprint, which was originally posted on Research Square and uses a slightly different template.

(3) Line numbers shown in revised manuscript.

(1) First sentence of introduction: "region economy" -> "region's economy"

(2) Edited.

(3) "The South and East Asian monsoon systems are of key importance for the region's economy, agriculture and industry."

(1) P14: a little bug: ¡60W/m2

(2) Latex encoding error.

(3) "At low (<60W/m$^2$) aerosol forcing"

(1) P14: form -> from: form 0 to 150W/m2

(2) Edited.

(3) "As the radiative forcing is increased from 0 to 150W/m$^2$"

(1) P14: sdepends -> depends: with 2xCO2andaerosol only, sdepends

(2) Already edited in submitted version.

(3) "*aerosol with 2xCO$_2$* and *aerosol only*, depends only"

---

## Author Comment (AC2)

**Response to reviewer 2**

Format of responses: (1) comments from referees/public, (2) author's response, and (3) author's changes in the manuscript.

**Specific comments:**

(1) Model evaluation against the observations is missing in the manuscript. Some meteorological parameters like precipitation, winds, and surface temperature as shown are needed to be validated against the observations.

(2) We agree with the reviewer that this should have been discussed in greater detail. Comparison to observations was done at an early stage but not all figures/analysis were included in the manuscript due to concerns over length. Given the idealistic nature of the experiments, extensive quantification of performance against, for example, satellite data, isn't applicable. Our requirements are that PLASIM reproduces the large-scale features of the monsoons, with a clear seasonal cycle, which is hopefully now evident.

(3) We have added subplots to Figures 1-4 to show the equivalent variables from ERA5 reanalysis data, to allow for visual comparison with the PLASIM model. Updated *"Section 1.2 Model Validation"* to reference and discuss the updated figures. Example of revised Figure 1 below.

[Figure]

(1) The values selected for aerosol forcings are too high. The values ranging from 30 to 40 W m$^{-2}$ are valid over a local region and season depending upon the emission type but applying all over India or eastern China could have an overestimation of aerosol effects. What is the rational thinking behind increasing aerosol forcing to 150 Wm$^{-2}$? If there is no interactive chemistry component in the model then these are highly idealized simulations. I suggest making it clear in the title.

(2) We have included additional figures for 30W/m$^2$ forcing as well as the 60W/m$^2$ forcing, to help quantify the linearity. Nonetheless, we remark that locally, in heavily polluted urban conglomerate and industrial regions, forcings of around 100 W/m$^2$ have been observed – see references in Section 2.2. In particular, see Table 2 in Kumar and Devara (2012), where values of -46 to -110/+46 to 115 W/m$^2$ are quoted for surface/atmosphere forcing in Delhi. As mentioned in the manuscript, the point of including unrealistically high forcings is to cover a parametric range of forcings. In particular, to see the behaviour leading up to a breakdown or severe weakening of the monsoon systems. The title mentions "an intermediate complexity climate model", and states that we are modelling the *effect* of absorbing aerosol forcing rather than explicitly adding aerosols. It is mentioned in Section 2.2 Experiment design that "The PLASIM model has no explicit treatment of aerosol interactions".
(3) We have added Figures in Section 5.1 to include results from simulations with an absorbing aerosol forcing of 30 W/m$^2$.

(1) I assume these values 30 to 150 Wm$^{-2}$ depict the aerosol atmospheric forcings (Top of the atmosphere (TOA) – surface) right? then why only it is applied to 550-750 hPa? Please clarify.
(2) We are mostly considering the case of absorbing aerosols. To a first approximation, the net energetic impact on the atmosphere column (top of atmosphere - surface) is zero. A heating of varying intensity, say H, is applied over 3 model levels which roughly correspond to 550-750 hPa (mid-troposphere). Thus, each of the 3 model levels has an applied forcing of intensity H/3. At the same time, a cooling of intensity -H is applied at the surface, because less solar radiation reaches the surface. This is now further clarified in the text.
(3) Added sentences in Section 2.2 to clarify that heating of intensity H/3 is applied at 3 mid-tropospheric levels, and also a cooling of intensity -H is applied at the surface.

(1) It is a bit confusing that if the mid-tropospheric heating is applied then the monsoon circulation at 850 hPa should have strengthened over India and southern China. In general, it should have created a mid-tropospheric temperature gradient but I see a consistent decrease in the precipitation. Please clarify.
(2) As mentioned above, the aerosol forcing consists of both a mid-tropospheric heating and a compensating surface cooling, such that the net change in the atmospheric column is zero. The absorbing aerosol forcing has a stabilising effect, increasing the stratification of the atmosphere and leading to reduced (convective) precipitation and weakening circulation (e.g. Li et al. (2016); Wilcox et al. (2020); Ayantika et al. (2021); Cao et al. (2022) – see manuscript for full references).
(3) No changes implemented.

(1) Could you please include some discussion on why there is an increase in precipitation over north India in 120 to 150 Wm$^{-2}$ aerosols forcing?
(2) As discussed in the text (Section 5), the overall effect of applying aerosol forcing over India is smaller than what is realised in the two other regions because aerosol forcing over East China leads to increasing precipitation over Northern India, thus countering the effect of the local forcing and making it non-statistically significant even for very strong forcing.
(3) Figures have been updated to show stippling where the anomaly is greater than 2x interannual variability: there is no stippling over North India so the increase in precipitation is much less significant than the reduction in precipitation across South India, Northeast India, East China & Southeast Asia.

(1) The responses in monsoon precipitation obtained in this paper could be possibly due to large amounts of scattering aerosols (anthropogenic sulfates) which seems consistent with the earlier published literature. Here, through absorbing aerosol forcings, a similar effect is obtained. Why? I am not sure whether the dynamics are correctly responding to aerosol forcing.

(2) We see a reduction in precipitation, a surface cooling and a weakening of the circulation, primarily in the regions where the absorbing aerosol forcing is applied. Thus, these results are consistent with published literature – see Section 1 for references. Some effects on the monsoons, such as surface cooling and a reduction in moisture availability, are common to both absorbing and scattering aerosols. Further work aims to isolate the specific effects of scattering aerosols, as opposed to absorbing aerosols, on the Asian monsoons, in a similar way to Herbert et al. 2022.
(3) No changes implemented.

(1) Whether the surface temperature anomalies or the tropospheric temperature anomalies induced by aerosol/2xCO$_2$ forcings are more sensitive in driving monsoon precipitation should be pointed out.
(2) The applied forcings affect both the surface temperature and the mid-troposphere. We do not see a strong reason to separate the effect of the two anomalies, as we are focusing on the forcing itself. In the case of the aerosol forcing, the surface and the mid-tropospheric anomalies both contribute to increasing the static stability of the atmosphere, thus decreasing the convective precipitation.
(3) No changes implemented.

(1) It would be nice to check the latitudinal cross-section of changes in air temperature vertically due to aerosol forcings. Subsequently, then while including 2xCO$_2$ forcings.
(2) We agree with the reviewer. Sections at 20°N and 109°E have been added to show the vertical temperature profile under absorbing aerosol forcing, with stippling to show statistically significant changes. Over the regions of applied forcing, a warm temperature anomaly can be seen around 500-850 hPa, and a cool temperature anomaly close to the land-surface. Similarly to Supplementary Figure S4, the *aerosol with 2xCO$_2$* simulation shows warmer temperatures, particularly at mid-levels, than the *aerosol* only simulation.
(3) Added figures for a longitudinal & a latitudinal cross section of temperature. Example below.

[Figure]

JJA: temperature (°C) at 109°E

JJA: temperature anomaly (°C) at 109°E

JJA: temperature anomaly (°C) at 109°E

(1) Inspection of vertical velocity responses to aerosol and 2xCO₂ forcings could be useful. Please check.

(2) We agree that this is a useful diagnostic, although convective precipitation is the meteorological field of greatest interest for us. We are including additional figures and analysis to include vertical velocity responses.

(3) Added extra figures of vertical velocity (omega) and updated the text to include analysis of added figures. Example vertical cross section of vertical velocity at 109°E shown below, with the dotted line showing convective precipitation. One can clearly see a strong reduction of convective motion and corresponding reduction of convective precipitation.

[Figure]

(1) "The intense surface cooling is primarily responsible for activating the ice-albedo effect.." on page 10. How does the ice-albedo affect the vertical distribution of temperature?
(2) As the surface temperatures cool to the extent that oceans begin to freeze, the increasing area of sea ice and snow cover leads to a higher albedo, thus increasing the amount of incoming radiation being reflected back into space and causing temperatures to drop. It is a positive feedback; as sea ice/snow cover increases, surface temperatures continue to decrease. This impacts the atmospheric layers above leading to greatly decreased temperature.
(3) Modified the relevant sentence: "The intense surface cooling is primarily responsible for activating the ice-albedo effect; a positive feedback which enhances surface cooling as sea ice and snow cover increases, causing a greater amount of radiation to be reflected back into space. The result is similar to the establishment of a nuclear winter, albeit via a slightly different mechanism."

(1) I could not understand how the aerosol forcing applied to eastern China increases precipitation over India. Are the changes statistically significant? There is still a decrease in surface temperature over India (Figure 11) without aerosol forcings. What could be the potential reason for an increase in precipitation? Please provide a physical explanation.
(2) The fact that the presence of an aerosol forcing over East China leads to an increase of precipitation over Northern India is an interesting phenomenon. Partly comparable results had been found in Herbert et al. 2022. Understanding this process is highly nontrivial and we are collaborating with the Herbert et al. team exactly to discover the mechanisms in action. Our initial theory is that although the absorbing aerosol forcing causes the low-level southwesterly wind in the band 0-20°N to weaken, there is a thin band around 20-25°N at high levels of forcing where the wind speed increases. We tend to attribute the increase in precipitation over North India to the increased wind speed, which brings an influx of moisture from the Arabian Sea.

Additionally, figures have been modified so that stippling indicates statistically significant changes, defined as where the change is greater than double the interannual June-July-August variability (standard deviation). Thus, although there is a slight increase in precipitation over North India, it is not statistically significant by our condition.
(3) Added stippling to figures (see revised Figure 11 below) and modified a sentence in Section 5: "The precipitation response of India to forcing applied over East China is nearly as strong as when the forcing is applied locally, albeit with opposing trends. Similar asymmetry in the teleconnection

between East China and India in relation to local absorbing aerosol forcing has been shown in Herbert et al. (2022); however, further study is required to fully understand the underlying mechanisms."

[Figure]

(1) The concept of the advection of dry air from Siberia is not well interpreted. It would be better to have a look at the responses of specific humidity to aerosol forcings.

(2) We agree with the reviewer that specific humidity is a key variable to consider. From Figure 6, we see a reduction in precipitable water over Southeast Asia and from Figure 7, an increase in 850 hPa wind speed from Southeast Asia to eastern Siberia. Thus, there is advection of dry air from Southeast Asia to eastern Siberia, leading to a reduction of precipitation in eastern Siberia. In support of this theory, there is a decrease in specific humidity at low levels over Southeast Asia and East China.  In particular, we see the specific humidity at 925 hPa decrease around 45°N, 125°E, which is outside the area of applied aerosol forcing.

(3) Added an extra figure to show specific humidity at 700 & 925 hPa (example below).

[Figure]

JJA: specific humidity (g/kg) at 925 hPa level

JJA: specific humidity anomaly (g/kg) at 925 hPa level

JJA: specific humidity (g/kg) at 700 hPa level

JJA: specific humidity anomaly (g/kg) at 700 hPa level

Technical comments:

(1) Maps of good quality are not well superimposed on the spatial figures. They seem at a very coarse resolution and distorted.

(2) We use a reasonably coarse horizontal resolution of approximately 2.8 degrees (T42 spectral resolution). This is the highest resolution available with the PLASIM model. The figures reflect the model resolution. The coastlines are drawn using the model's land-sea map, so as to accurately reflect the horizontal resolution of the model and not be misleading. For future work, it is hoped that similar experiments with a gradually varying forcing might be performed with the WRF model (or similar).

(3) No changes implemented.

---

## Author Comment (AC3)

**Response to reviewer 3**

Format of responses: (1) comments from referees/public, (2) author's response, and (3) author's changes in the manuscript."

**General comments:**

(1) The manuscript describes the use of an intermediate complexity model to perform sensitivity experiments to prescribed aerosols and CO2 forcing over Asia, in combination as well as separately, also distinguishing between sub-regional forcing and identifying non-linearities.

The topic has been widely studied using a variety of modelling tools and simulations. Needless to say, there are still uncertainties and varied responses across models. It is a very important topic as aerosols are still very high over the region studies, and there may be large differences in the spatio-temporal evolution of regional emissions in the coming decades. The manuscript is well written and clear. I have some concerns on some of the experimental set-up and analysis that need to be addressed before acceptance.

(2) We thank the reviewer for their comments and for taking the time to review the manuscript.

**Major comments:**

(1) The model validation should be done by comparing the output to observations.

(2) We agree with the reviewer that this should have been discussed in greater detail. Comparison to observations was done at an early stage but not all figures/analysis were included in the manuscript due to concerns over length. Given the idealistic nature of the experiments, extensive quantification of performance against, for example, satellite data, isn't applicable. Our requirements are that PLASIM reproduces the large-scale features of the monsoons, with a clear seasonal cycle, which is hopefully now evident.

(3) We have added subplots to Figures 1-4 to show the equivalent variables from ERA5 reanalysis data, to allow for visual comparison with the PLASIM model. Updated *"Section 1.2 Model Validation"* to reference and discuss the updated figures. Example of revised Figure 1 below.

[Figure]

(1) The magnitude of radiative forcing is definitely too large. Available estimates provide values up to 20/30 W m2 at the surface, I think it is generally OK to use larger values than observed in idealised settings (e.g., PDRMIP) to highlight the signal, but the values used here are definitely too large. I think these simulations are unrealistic.

(2) We have included additional figures for 30W/m$^2$ forcing as well as the 60W/m$^2$ forcing, to help quantify the linearity. Nonetheless, we remark that locally, in heavily polluted urban conglomerate and industrial regions, forcings of around 100 W/m$^2$ have been observed – see references in Section 2.2. In particular, see Table 2 in Kumar and Devara (2012), where values of -46 to -110/+46 to 115 W/m$^2$ are quoted for surface/atmosphere forcing in Delhi. As mentioned in the manuscript, the point of including unrealistically high forcings is to cover a parametric range of forcings. In particular, to see the behaviour leading up to a breakdown or severe weakening of the monsoon systems.

We use an intermediate complexity climate model to help bridge the gap between conceptual (e.g. box) and advanced climate models (i.e. CMIP6 standard). For our simulations, we do not require the same level of realism that CMIP6 standard models aim to provide, because we are not trying to predict a future climate state or contribute to policy guidance on climate change. The benefits of using an intermediate rather than an advanced complexity climate model are highlighted in Section 2. For future work, it is hoped that similar experiments with a gradually varying forcing might be performed with a higher complexity model, such as the WRF model.

(3) We have added Figures in Section 5.1 to include results from simulations with an absorbing aerosol forcing of 30 W/m$^2$. Additional sentence added to Conclusions: "Our results are limited by the low resolution and lack of explicit aerosol interactions and chemistry, but future work will aim to address these issues by repeating similar experiments using a more complex global climate model."

(1) Along these lines, I am not sure the comparison with 2xCO2 is appropriate, and especially the identification of which of the two drivers dominates.

(2) At present, $2xCO_2$ is in some sense unrealistic; we are fortunately still far from a 720 ppm CO@ global $CO_2$ concentration. However, there are many modelling studies that use the standard IPCC forcing scenarios, which include extreme storylines. Indeed, it is typically the most extreme scenarios (i.e. RCP8.5/SSP5-8.5) that are most often represented in the literature. From IPCC6, two out of the four climate scenarios result in double or greater levels of carbon dioxide levels by 2100, making $2xCO_2$ a reasonable choice when considering future greenhouse gas forcing in high to extreme scenarios. The relative dominance of greenhouse gas and aerosol forcings in the actual future scenarios depends on the path of change of such forcings, which is associated with a high degree of uncertainty. The goal is to compare a reference climatic forcing ($2xCO_2$) with different scenarios of aerosol forcing.

The $2xCO_2$ simulations are performed with absorbing aerosol forcing from 0 to 150W/m$^2$, so can we consider the impact of doubling carbon dioxide levels in tandem with a range of different intensity absorbing aerosol forcing. We find a fair degree of linearity in the response to aerosol forcings of 30, 60 & 90W/m$^2$ with $2xCO_2$, which can be seen by the similarity in left to right columns of Figures 9 & 10. In our experiments, we find that the system is more sensitive to absorbing aerosol than carbon dioxide forcing.

(3) As per previous comment, we have added Figures to show the sensitivity to both 60 W/m$^2$ and 30 W/m$^2$ absorbing aerosol forcing. Added sentences to the Abstract ("These results suggest that in the future, the anticipated reduction in aerosol concentration may have a greater impact on monsoonal precipitation than the increase in greenhouse gases.") and Conclusions ("We find that the precipitation response is more sensitive to absorbing aerosol than greenhouse gas forcing, highlighting the importance of air quality policies and the impact they can have on the future state of the South and East Asian monsoons.")

(1) Also, the prescribed forcing simulates the effect of absorbing aerosols, rather than sulfate. In reality, the latter are found to dominate the aerosol-driven monsoon changes. I think it is important to further underscore these differences and make it clearer when drawing conclusions.

(2) We agree with the reviewer (and with reviewer 1) that this needs to be clarified. The manuscript has been revised to be clear that the forcing used is representative of absorbing aerosol effects.

Scattering and absorbing aerosols and their impact on the South and East Asian monsoons are discussed in the Introduction. Although several articles suggest that sulphates are the dominant aerosol species in terms of the effect on monsoonal rainfall, others suggest that black carbon has the greater effect. There remains much debate and considerable uncertainty on the effects of different aerosol species and their effect on the vertical profile of the atmosphere. The location and spatial pattern of aerosol loading is also important, and can have contrasting effects on precipitation in the local region compared to a remote region. One of our aims for future work is to repeat the experiments with a sulphate-style forcing, following Herbert et el. (2022), to compare the responses of the monsoons to different aerosol forcings.

(3) Changed various instances of "aerosols" to "absorbing aerosols".

(1) As there is not seasonality in the forcing here, can the authors speculate on what this means in terms of realism of summer anomalies, also in terms of preconditioned conditions through the previous winter and spring?

(2) PLASIM's ability to represent long-term (monthly and greater) correlations in the soil properties is limited, because of the simplified nature of the land module. Hence, we expect that whether we use a perennial aerosol forcing or an aerosol forcing that is active each year for a period that includes the summer season, the June-July-August cumulative precipitation should be approximately

the same. We plan to investigate this more in detail in the forthcoming study in collaboration with the Herbert et al. 2022 authors.

(3) Added a sentence to the conclusions regarding our intentions to investigate this matter further: "we will consider the impact of scattering aerosol forcing compared with absorbing aerosol forcing, and the effect of applying forcing over a shorter seasonal period, rather than perennially."

(1) I suggest also making the text a bit more concise, as I think some parts are not needed. For example, all the section on tipping points in the introduction is not really necessary. Similarly, Section 4.1 is not contribution much to the overall discussion. There are also other parts throughout the manuscript that can be shortened.

(2) The Indian monsoon has long been considered a tipping element and one of our aims was to establish the level of forcing required to cause a transition or breakdown of the monsoon system, such that the summer precipitation becomes very low. It seems relevant to put our results and the problem of the response of the system to perturbations in the context of tipping points and of the literature of safe operating spaces.

We have tried hard to make the manuscript as compact as possible and we will make a further effort in the revised version, without risking the loss of key information relating to justification of methods and analysis of results.

(3) Section 4.1 has been modified to include some more quantitative analysis, following the comments of reviewer 1, increasing the value of this section.

(1) Why is relative humidity used instead of specific humidity? I do not think RH at 200 hPa (Fig. 3) is really useful. The way winds are plotted is rather unusual (scale the arrows by the magnitude instead of plotting magnitude and direction separately). Also, it may be useful to calculate vertical integrated moisture transport.

(2) At 850 hPa, relative humidity seems relevant because we wish to understand where we are closer to saturation, and hence closer to conditions conducive to convective precipitation. At 200 hPa, where the temperature is extremely low, specific humidity is virtually zero by the Clausius-Clapeyron relationship. Thus, in our opinion, plotting specific humidity at 200 hPa provides very little information. However, we agree that specific humidity is an important variable to consider and we have added a figure to show the specific humidity at 925 and 700 hPa.

For Figures 2 & 3, wind vectors have been scaled by magnitude, to allow for plotting over coloured contours of relative humidity. This reduces the number of overall figures and shortens the manuscript, without losing visual representation of key variables. In Figure 7, the wind speed and direction are plotted separately, so that we can more clearly focus on the changes in wind speed.

With the additional figures of specific humidity and vertical velocity (following reviewer suggestions), we can infer moisture advection horizontally and vertically. With concerns over length, we feel that another figure to show vertically integrated moisture transport is superfluous.

(3) Added extra figures of specific humidity (shown below) and vertical velocity (omega), and updated the text to include analysis of added figures. Example vertical cross section of vertical velocity at 109°E shown below, with the dotted line showing convective precipitation.

[Figure]

[Figure]

(1) Differences should have plotted a corresponding statistical significance.

(2) We agree with the reviewer. We have updated the figures to show stippling where the changes are statistically significant. Here, statistical significance is determined by where the anomaly (e.g. $60W/m^2$ forcing – control) is greater than 2 times the interannual variability of the June-July-August mean, with respect to the reference simulation (e.g. $60W/m^2$ forcing).

(3) Figures have been updated with stippling to show statistically significant changes. In conjunction, the text has edited to include discussion of the statistical relevance of the anomaly signals. Example of figures with added stippling can be seen in response to some of the comments above.

(1) The analysis of regional responses and the reciprocal influence between India and China is, in my opinion, the most interesting part. A dynamical analysis of the upper level circulation will also be useful.

(2) The fact that the presence of an aerosol forcing over East China leads to an increase of precipitation over Northern India is an interesting phenomenon. Partly comparable results had been found in Herbert et al. 2022. Understanding this process is highly nontrivial and we are collaborating with the Herbert et al. team exactly to discover the mechanisms in action.

Section 3 includes some analysis regarding the upper level circulation, but we have extended this to include discussion of vertical velocity and specific humidity, so that horizontal and vertical transport of moisture can be considered in a holistic way.

(3) Analysis of results in Sections 3 & 4 has been extended to include discussion of specific humidity and vertical velocity. Added stippling to figures (see revised Figure 11 below) and modified a sentence in Section 5: "The precipitation response of India to forcing applied over East China is nearly as strong as when the forcing is applied locally, albeit with opposing trends. Similar asymmetry in the teleconnection between East China and India in relation to local absorbing aerosol forcing has been shown in Herbert et al. (2022); however, further is required to fully understand the underlying mechanisms."

(1) What are the limitation of this study and in particular of using PLASIM? This should be clearly stated in the conclusions.

(2) We agree with the reviewer that it is important to consider the caveats of using an idealised modelling scenario, which we have discussed in detail in Section 2.2. We have also added a sentence to the Conclusions (see below).

(3) Following this comment and comment by reviewer 1, we have added a sentence to the Conclusions to highlight the limitations of our study, and our future work aims: "Our results are limited by the low resolution and lack of explicit aerosol interactions and chemistry, but future work will aim to address these issues by repeating similar experiments using a more complex global climate model."

---

## Author Response (AR2)

**Second response to reviewer 3**

Format of responses: (1) comments from referees/public, (2) author's response, and (3) author's changes in the manuscript."

**Minor comments:**

(1) In one of my comments, I meant vertically-integrated moisture transport, which is a lat-lon quantity, not vertical transport.

(2) Following on from the original comment regarding figure 3, which shows the 200 hPa relative humidity and circulation for the PLASIM model and the ERA5 reanalysis dataset, we have included another figure in the supplementary material to show the vertically integrated moisture flux (& vertically integrated moisture flux convergence) of both PLASIM and ERA5 for summer and winter seasons.

(3) Included supplementary figure (S1 – shown below) of the vertically integrated moisture flux (& vertically integrated moisture flux convergence) for PLASIM and ERA5, for summer and winter seasons. Updated text in section 2.1 Model Validation to include reference to the new figure.

[Figure]

(1) It looks like the choice of 109E to plot meridional cross sections is not optimal, considering the vicinity to the TP topography (I see some of the effects in the horizonal maps). Also, generally one plots averages over a longitudinal band.

(2) We agree with the reviewer and have updated all sections taken at 109°E to 115°E. Moving much farther east would mean that the section is mostly taken over ocean points, where no aerosol forcing is applied, and hence wouldn't be as relevant. Sections have been considered in the range 105-120°E but they are sufficiently similar that only one longitudinal section per variable is shown as an example.
(3) Sections (supplementary figures S5-7 & S10-12) changed from 109°E to 115°E.